# Efficient targeted integration directed by short homology in zebrafish and mammalian cells

Wesley A Wierson[1†], Jordan M Welker[1†], Maira P Almeida[1†], Carla M Mann[1], Dennis A Webster[2], Melanie E Torrie[1], Trevor J Weiss[1], Sekhar Kambakam[1], Macy K Vollbrecht[2], Merrina Lan[1], Kenna C McKeighan[1], Jacklyn Levey[1], Zhitao Ming[1], Alec Wehmeier[1], Christopher S Mikelson[1], Jeffrey A Haltom[1], Kristen M Kwan[3], Chi-Bin Chien[4], Darius Balciunas[5], Stephen C Ekker[6], Karl J Clark[6], Beau R Webber[7], Branden S Moriarity[7], Stacy L Solin[2], Daniel F Carlson[2], Drena L Dobbs[1], Maura McGrail[1*], Jeffrey Essner[1*]

[1]Department of Genetics, Development and Cell Biology, Iowa State University, Ames, United States; [2]Recombinetics, Inc, St. Paul, United States; [3]Department of Human Genetics, University of Utah School of Medicine, Salt Lake City, United States; [4]Department of Neurobiology and Anatomy, University of Utah Medical Center, Salt Lake City, United States; [5]Department of Biology, Temple University, Philadelphia, United States; [6]Department of Biochemistry and Molecular Biology, Mayo Clinic, Rochester, United States; [7]Department of Pediatrics, Masonic Cancer Center, University of Minnesota, Minneapolis, United States

*For correspondence:
mmcgrail@iastate.edu (MMG);
jessner@iastate.edu (JE)

†These authors contributed equally to this work

**Abstract** Efficient precision genome engineering requires high frequency and specificity of integration at the genomic target site. Here, we describe a set of resources to streamline reporter gene knock-ins in zebrafish and demonstrate the broader utility of the method in mammalian cells. Our approach uses short homology of 24–48 bp to drive targeted integration of DNA reporter cassettes by homology-mediated end joining (HMEJ) at high frequency at a double strand break in the targeted gene. Our vector series, pGTag (plasmids for Gene Tagging), contains reporters flanked by a universal CRISPR sgRNA sequence which enables in vivo liberation of the homology arms. We observed high rates of germline transmission (22–100%) for targeted knock-ins at eight zebrafish loci and efficient integration at safe harbor loci in porcine and human cells. Our system provides a straightforward and cost-effective approach for high efficiency gene targeting applications in CRISPR and TALEN compatible systems.

## Introduction

Designer nucleases have rapidly expanded the way in which researchers can utilize endogenous DNA repair mechanisms for creating gene knock-outs, reporter gene knock-ins, gene deletions, single nucleotide polymorphisms, and epitope tagged alleles in diverse species (*Bedell et al., 2012*; *Beumer et al., 2008*; *Carlson et al., 2012*; *Geurts et al., 2009*; *Yang et al., 2013*). A single dsDNA break in the genome results in increased frequencies of recombination and promotes integration of homologous recombination (HR)-based vectors (*Hasty et al., 1991*; *Hoshijima et al., 2016*; *Orr-Weaver et al., 1981*; *Rong and Golic, 2000*; *Shin et al., 2014*; *Zu et al., 2013*). Additionally, in vitro or in vivo linearization of targeting vectors stimulates homology-directed repair (HDR) (*Hasty et al., 1991*; *Hoshijima et al., 2016*; *Orr-Weaver et al., 1981*; *Rong and Golic, 2000*; *Shin et al., 2014*; *Zu et al., 2013*). Utilizing HDR or HR at a targeted double-strand break (DSB) allows directional

knock-in of exogenous DNA with base pair precision, however, reported frequencies vary widely, and engineering targeting vectors with long homology arms is not straightforward.

Previous work has shown *Xenopus* oocytes have the ability to join or recombine linear DNA molecules that contain short regions of homology at their ends, and this activity is likely mediated by exonuclease activity allowing base pairing of the resected homology (*Grzesiuk and Carroll, 1987*). More recently, it was shown in *Xenopus*, silkworm, zebrafish, and mouse cells that a plasmid donor containing short (≤40 bp) regions of homology to a genomic target site can promote precise integration at the genomic cut site when the donor plasmid is cut adjacent to the homology (*Aida et al., 2016*; *Hisano et al., 2015*; *Nakade et al., 2014*). Gene targeting in these studies is likely mediated by the alternative-end joining/microhomology-mediated end joining (MMEJ) pathway or by a single strand annealing (SSA) mechanism (*Ceccaldi et al., 2016*), collectively referred to as a homology mediated end joining (HMEJ). In contrast, in human cell culture, linear donors using a similar strategy with homologous ends have been reported to show inefficient integration until homology domains reach ~600 bp (*Zhang et al., 2017*), suggesting different repair pathways may predominate depending on cell type. In the initial reports using short regions of homology for in vivo gene targeting in zebrafish, the level of mosaicism in F0 injected animals was high, resulting in inefficient recovery of targeted alleles through the germline (*Aida et al., 2016*; *Hisano et al., 2015*; *Luo et al., 2018*; *Nakade et al., 2014*). Most recently, studies in *Drosophila* show efficient integration of exogenous DNA in flies and S2 cells using 100 bp homology arms flanked by a CRISPR target site for in vivo homology liberation (*Kanca et al., 2019*). Together, these studies suggest a strategy that combines short homology flanked donors with in vivo homology arm liberation should lead to efficient precision targeting in zebrafish and mammalian cells.

Here, we present GeneWeld, a HMEJ strategy for targeted integration directed by short homology and demonstrate efficient germline transmission rates for recovery of targeted alleles in zebrafish. We provide a suite of donor vectors, called pGTag, that can be easily engineered with homologous sequences (hereafter called homology arms) to a gene of interest and a web interface for designing homology arms (www.genesculpt.org/gtaghd/). We demonstrate that 24 or 48 base pairs of homology directly flanking cargo DNA promotes efficient gene targeting in zebrafish, pig, and human cells. Our results also suggest that longer homology arms up to 1 kb in length do not increase the frequency of on-target integration in zebrafish in comparison to 24 or 48 bp. Our experiments illustrate that Cas9 mediated in vivo release to expose homology on the knock-in cassette ends is integral for enhancing targeted integration. Using short homology-arm mediated end joining, we can achieve germline transmission rates averaging approximately 50% across several zebrafish loci when pre-selecting reporter-positive embryos to raise to adulthood. Southern blot analysis in the F1 generation reveals we can recover single copy integration alleles with precision at both 5' and 3' ends at high frequency, enabling efficient recovery of zebrafish with precise site-directed gene modifications. We present a strategy to delete and replace up to 48 kb of genomic DNA with a donor containing homology arms flanking two distal CRISPR/Cas9 sites in a gene. Finally, we show that short homology arm-directed targeted integration of a GFP reporter into safe harbor loci in porcine and human cells increased 10-fold in comparison to homologous recombination. The tools and methodology described here provide a tractable solution to creating precise targeted integrations and open the door for other genome editing strategies using short homology.

## Results

### A single short homology domain drives efficient CRISPR targeted integration

To develop baseline gene targeting data, we engineered variable length homology domains to target *noto,* using a similar strategy to *Hisano et al. (2015)*. Homology lengths were based on observations that DNA repair enzymes bind DNA and search for homology in 3 or 4 base pair lengths (*Figure 1a*; *Conway et al., 2004*; *Singleton et al., 2002*). We engineered a 2A-TagRFP-CAAX-SV40 donor vector to contain a sgRNA site that matches a site in exon 1 of the zebrafish *noto* gene. The donor vector used a 2A peptide sequence in frame with *noto* and TagRFP to allow separation of the fluorescent protein from the nascent protein of the disrupted *noto* gene during translation. For fluorescence to be detected, the integrated reported gene is required to be in frame with the open

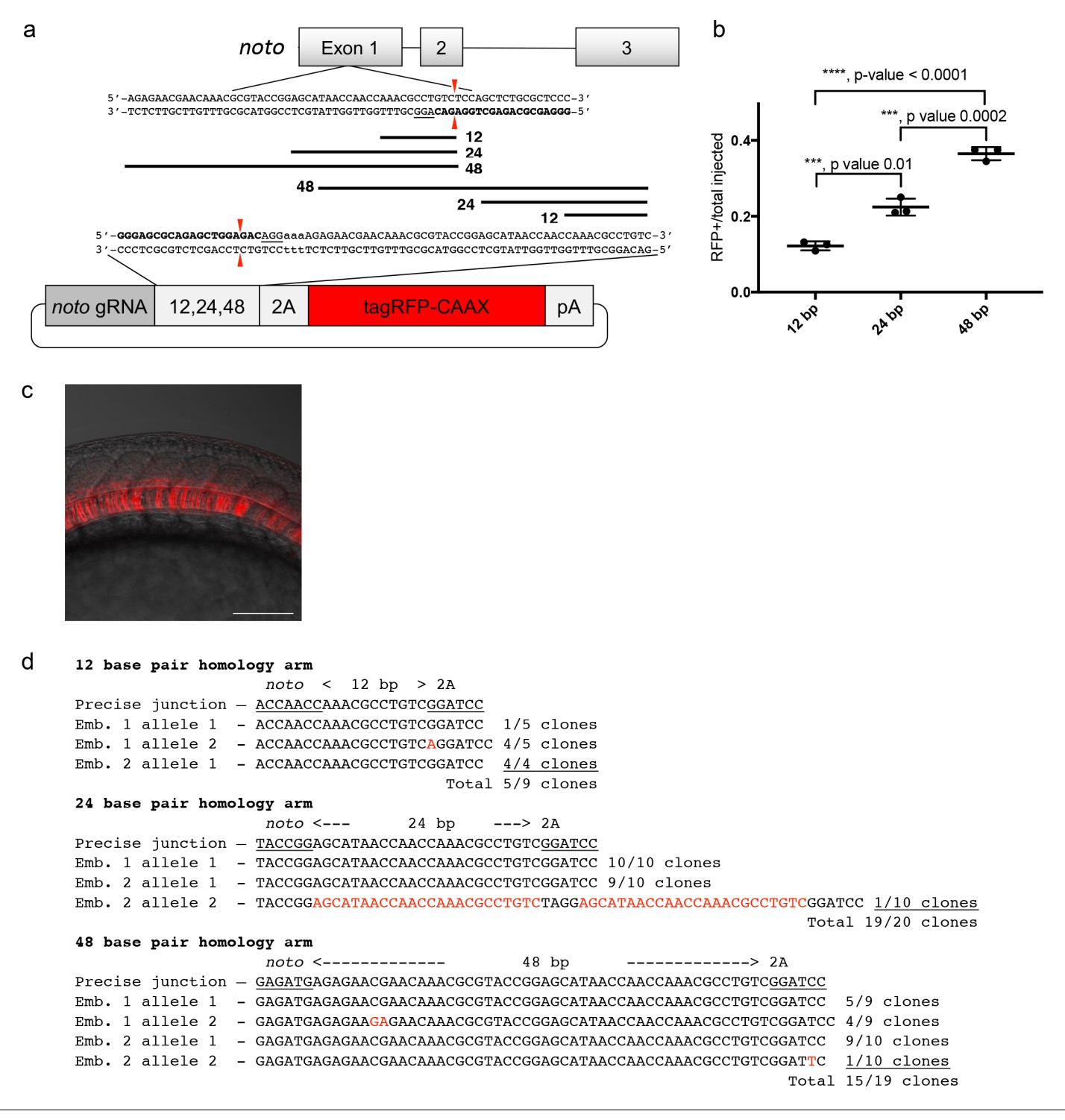

**Figure 1.** A single short homology arm 5' to the sgRNA target site in the *noto* gene targets integration in zebrafish embryos. (**a**) Schematic for *noto* homology arm and donor vector design. Bold letters show the *noto* sgRNA target sequence in the genome. This sgRNA target sequence was also used to target Cas9 cutting in the donor vector. Black bars represent the different homology arm lengths 12, 24, or 48 bp, used to target the 2A-tagRFP-CAAX donor vector into the *noto* exon 1 target site. PAM sequences are underlined. Red arrows indicate the Cas9 cut site 3 bp upstream of the PAM. The 3 nucleotide spacer lacking homology to the genome is represented by the lowercase sequence 'aaa' located in between the donor vector PAM and the 5' end of the homology arm. (**b**) Targeting efficiency of *noto* exon1 2A-tagRFP-CAAX donor vectors containing a single 5' homology arm of 12, 24, or 48 bp. Data represents mean +/- s.e.m. of 3 independent targeting experiments. p values calculated using two-tailed unpaired t-test. (**c**) Live confocal image of *noto-2A-TagRFP-CAAX-SV40* targeted embryo showing specific RFP expression in the notochord. Scale bar, 100 μm. (**d**) Sanger

*Figure 1 continued on next page*

*Figure 1 continued*
sequencing of cloned 5' junction fragments from RFP positive F0 embryos, aligned to the expected sequence from a precise integration event. Numerator represents correct clones, denominator represents total clones sequenced. Junctions are considered precise if the homology arm does not contain any mismatch and there are no insertions or deletions up- or downstream of the programmed homology.

The online version of this article includes the following figure supplement(s) for figure 1:

**Figure supplement 1.** The Universal sgRNA (UgRNA) promotes high efficiency targeted integration.

reading frame of *noto*. The sgRNA site was followed by 12, 24 or 48 bp of sequence homologous to the 5' region directly adjacent to the Cas9 cut site in *noto* exon 1 (*Figure 1a*). The *noto* exon 1 sgRNA targeting efficiency was tested by co-injection with Cas9 mRNA into zebrafish embryos. At 3 dpf, 5 larvae were pooled for genomic DNA extraction and PCR to generate a *noto* exon 1 amplicon. MiSeq analysis showed 95% of the amplified alleles contained indels at the *noto* sgRNA genomic target site (*Supplementary file 1* Table S1-S2). Injection into the cytoplasm of the 1 cell stage embryo of the donor cargo DNA together with *noto* exon 1 sgRNA and Cas9 mRNA resulted in efficient targeted integration, as observed by notochord-specific RFP expression and PCR amplification of junction fragments between the *noto* gene and the targeting donor (*Figure 1a-c*; *Supplementary file 1* Table S2-S3). The frequency of embryos with notochord-specific RFP expression increased with the length of the homology arm up to 48 bp (*Figure 1b*), suggesting that longer homology arms promote increased precise integration of the targeting construct.

To examine the targeted alleles, we performed junction fragment analysis by PCR amplification of the junction between the *noto* gene and the targeted construct. Junction fragment analysis in the RFP expressing embryos injected with the 12 bp homology arm construct revealed precise integration at the 5' end in 56% of the sequenced alleles (5/9 junctions) (*Figure 1d*). The imprecise allele was a single base insertion between the homology domain and PAM that creates a frame shift and would not produce RFP expression. 95% of the recovered alleles following injection with the 24 bp homology arm construct (19/20 junctions) were precise, with the one imprecise allele containing a duplication of the homology domain. This latter allele most likely occurred by non-homologous end joining (NHEJ) rather than homology directed repair. 79% of the recovered alleles from embryos injected with 48 bp homology arm construct (15/19 junctions) were also precise (*Figure 1d*). In these experiments we did not analyze the 3' end of the integration, since the targeting construct only had a 5' homology arm and the 3' junctions are likely resolved by the NHEJ pathway, similar to previous reports (*Auer et al., 2014*; *Maresca et al., 2013*; *Suzuki et al., 2016*). Taken together, these experiments suggested that short homology arms can promote precise integration of a targeting construct by homology directed repair at relatively high frequency, however, imprecise alleles containing single base substitutions and insertions generated by NHEJ are also recovered.

## A universal guide RNA to liberate donor homology for targeted integration

To simplify targeting construct design and allow for consistent, reproducible liberation of the donor cargo in vivo by Cas9, we previously designed a universal guide RNA sequence, UgRNA (*Wierson et al., 2019a*), with optimal base composition using CRISPRScan (*Figure 1—figure supplement 1a*; *Moreno-Mateos et al., 2015*). The UgRNA does not have predicted targets in the zebrafish, pig, or human genomes, and in vivo use shows efficient double strand break induction and homology mediated repair at a target site in a fluorescent reporter integrated into the zebrafish *noto* gene (*Wierson et al., 2019a*). The UgRNA and a CGG PAM sequence were cloned 5' to the 24 bp *noto* homology arm in the 2A-TagRFP-CAAX-SV40 donor vector (*Figure 1—figure supplement 1b*). To test the ability of the UgRNA guide to direct a Cas9 double strand break to the vector in vivo and promote efficient targeted integration, zebrafish embryos were co-injected with Cas9 mRNA, UgRNA, *noto* sgRNA, and the *noto* targeting construct. 21% of injected embryos showed notochord-specific RFP expression, suggesting that Cas9 cutting at the vector UgRNA site efficiently exposes the donor vector 5' homology arm and drives precise targeted integration (*Figure 1—figure supplement 1c*, *Supplementary file 1* Tables S2-S3). This frequency was similar to the above experiments that used the gene specific *noto* sgRNA to cut the targeting construct.

## Dual homology arm liberation directs precise 5' and 3' integration in somatic tissue

We leveraged the activity of the UgRNA to develop GeneWeld, a strategy for targeted integration that promotes high frequency precision integration at both 5' and 3' junctions of the genomic target site. The donor vector contains 5' and 3' homology arms cloned on either side of the targeting cassette, which are flanked by UgRNA sites (*Figure 2a*) . The strategy takes advantage of DNA ends cut by CRISPR/Cas9 to initiate targeted integration directed by short homology (*Figure 2a*). A high efficiency nuclease introduces a DSB in the chromosomal target, and a second nuclease makes two DSBs in the vector at the UgRNA sites, exposing both 5' and 3' short homology arms. The complementarity between the chromosomal DSB and the donor 5' and 3' homology arms likely activates a MMEJ/SSA or other non-NHEJ DNA repair mechanism, together referred to as HMEJ. The reagents needed for this gene targeting strategy include *Cas9* mRNA to express the Cas9 nuclease, a guide RNA targeting the genomic sequence of interest, the universal sgRNA UgRNA, and the UgRNA donor vector with 5' and 3' homology arms complementary to the genomic target site (*Figure 2a*).

We built a series of vectors, pGTag, which contain sites on both sides of the cargo for cloning a short homology arm that is complementary to the 5' or 3' sequence flanking the genomic target site. The vectors also include the UgRNA sequence outside the sites for homology arm cloning (*Figure 2a, b*). A 3 bp spacer that lacks homology to the nucleotides that sit 5' to the start of the homology arm in the genome was added between the universal sgRNA PAM CGGand the homology arm (*Figure 2c*), in order to maintain the length of the homology arm at 24 or 48 bp. The 5' and 3' homology arms can be cloned simultaneously into the vector in a 1-pot reaction using type II restriction enzyme sites, or can be cloned sequentially. The final donor targeting vector contains a cargo flanked by 5' and 3' homology arms with UgRNA sequences on both ends. Following injection, we hypothesize that cleavage by Cas9 at the UgRNA sites liberates the DNA cargo from the plasmid

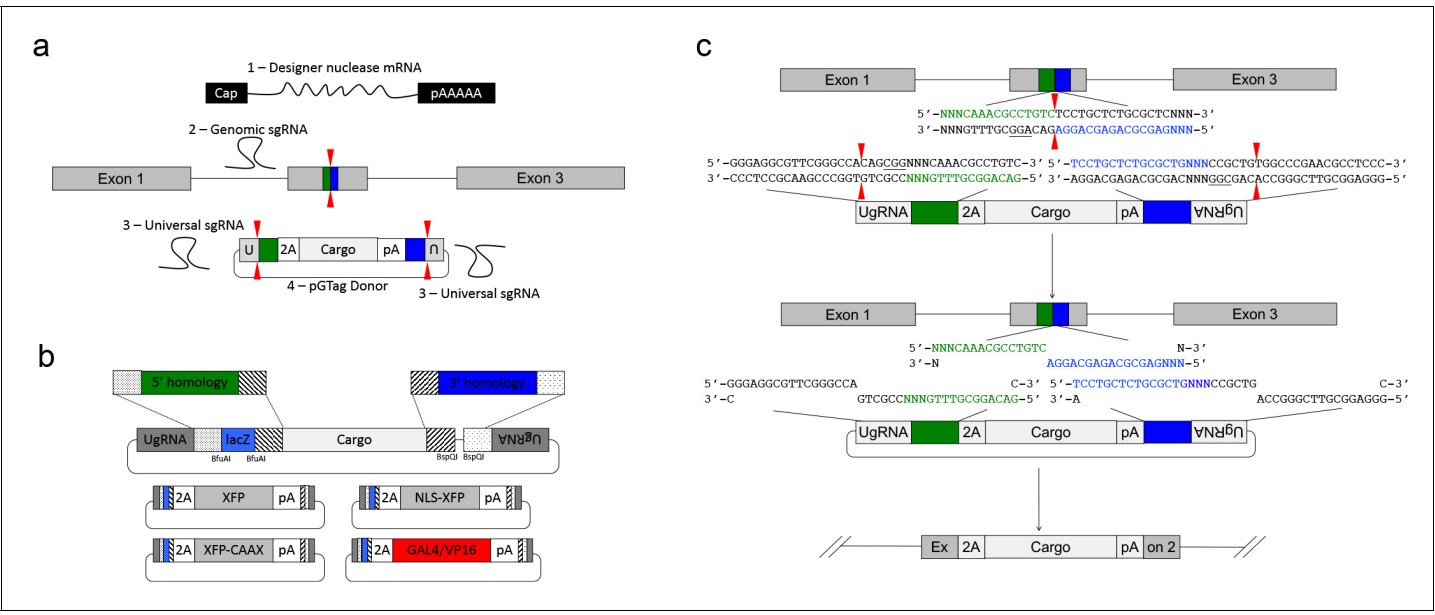

**Figure 2.** GeneWeld strategy and pGTag vector series. (**a**) GeneWeld reagent components are designed for simultaneous nuclease targeting of genome and donor to reveal short regions of homology. Red arrowheads represent nuclease DSB cut sites. Components include: 1 - Designer nuclease mRNA, either Cas9 to target both the genome and donor, or Cas9 to target the donor and TALEN to cut the genome; 2 - sgRNA for targeting Cas9 to genome; 3 - Universal sgRNA to liberate donor cargo and homologous ends; and 4 - pGTag donor of interest with short homology arms. (**b**) Stippled and striped boxes represent sticky ends created by Type IIs restriction endonucleases *Bfu*AI and *Bsp*QI, allowing digestion and ligation of both homology arms into the donor vector in a single reaction. Homology arm fragments are formed by annealing complementary oligonucleotides to form dsDNA with sticky ends for directional cloning into the vector. XFP = Green or Red Fluorescent Protein. pA = SV40 or β-actin 3' untranslated region. Red and green fluorescent proteins were cloned into the pGTag vectors, and for each color, subcellular localization sequences for either nuclear localization (NLSs) and membrane localization (CAAX) are provided. (**c**) Schematic of GeneWeld targeting in vivo. After designer nuclease creates targeted double-strand breaks in the genome and donor, end resection likely precedes homology recognition and strand annealing, leading to integration of the donor without vector backbone.

backbone and exposes both 5' and 3' donor homology arms for interaction with DNA on either side of the genomic DSB (*Figure 2c*).

To test this strategy, we targeted four genes in zebrafish, *notochord homeobox (noto)*, *tyrosinase (tyr)*, *endothelial cell adhesion molecule a (esama)*, and *connexin 43.4 (cx43.4)*, with efficient gRNAs which either produced a mutant phenotype or displayed 80–96% indel formation (*Supplementary file 1* Tables S1-S2). The frequency of expression from the pGTag targeting vector reporter genes was measured in somatic tissue following injection (*Figure 3a-d*). Injection of 24 or 48 bp homology arm *noto* 2A-eGFP-SV40 donors resulted in 24% of zebrafish embryos showing extensive reporter expression in the notochord (*Figure 3a,e*), indicating a similar in frame and precise integration efficiency compared to targeting with the single 5' homology arm 2A-TagRFP-CAAX-SV40 vector (*Supplementary file 1* Tables S2-S3). The results also suggest that when using the UgRNA to liberate the cargo, 24 bp of homology directs targeted integration as efficiently as 48 bp, further reducing the cost of homology arms for construction of targeting vectors.

Targeting exon 4 of *tyr* or exon 2 of *esama* with a 24 bp homology arm 2A-TagRFP-CAAX-SV40 donor did not result in detectable RFP signal in pigmented cells where *tyr* is expressed, similar to previous reports for *tyr* (*Hisano et al., 2015*). However, PCR amplification and sequencing of a fragment of the expected size that spans the exon-integration cassette junction from injected embryos indicated the 2A-RFP cassette was precisely integrating in frame in *tyr* exon 4 (*Figure 3—figure supplement 1*). This suggested the level of RFP expression was below the threshold of detection. To amplify the fluorescent signal, we built pGTag 24 bp homology arm 2A-Gal4/VP16-$\beta$−actin3'UTR targeting vectors with 24 bp homology arms to integrate the Gal4/VP16 trans-activator into the *tyr* and *esama* target sites. The Gal4 vectors were injected into transgenic zebrafish embryos carrying a 14xUAS-RFP reporter, *Tg(UAS:mRFP)*[tpl2] (*Balciuniene et al., 2013*). This resulted in strong RFP signal in 64% of *tyr* injected animals (*Figure 3b,e*), however, the embryos were highly mosaic, with only 9% of embryos displaying extensive RFP expression throughout most of the pigmented cells. Targeting *esama* exon 2 with 2A-Gal4/VP16 in the *Tg(UAS:mRFP)*[tpl2] transgenic background resulted in 21% of embryos displaying extensive RFP expression specifically in the vasculature where *esama* is expressed (*Figure 3c,e*). This approach was extended to five additional loci, targeting 2A-Gal4/VP16 to *filamin a (flna)* exon 4, *moesin a (msna)* exon 2 and 6, *aquaporin 1a1 (aqp1a1)* exon 1, *aquaporin 8a1 (aqp8a1)* exon 1, and *annexin a2a (anxa2a)* exon 3. At these loci, mosaic expression of RFP was observed following injection in 4–55% of *Tg(UAS:mRFP)*[tpl2] embryos in domains consistent with the normal expression of these genes (*Supplementary file 1* Table S2 and S3). Taken together, these results suggest that our short homology arm targeting strategy promotes high efficiency integration in somatic tissue and allows endogenous gene expression patterns to be followed in living embryos.

Previous work in zebrafish indicated that longer homology arms (200 bp to 1 kb in length), in combination with restriction enzyme digestion either in vitro or in vivo to liberate a linear donor template, could promote efficient targeted integration (*Hoshijima et al., 2016*; *Shin et al., 2014*). Using the pGTag vectors with UgRNA sites for in vivo homology arm liberation, we tested whether 1 kb long homology arms altered the efficiency of integration in comparison with 24 or 48 bp of short homology. Targeted integration of a 2A-TagRFP-CAAX-SV40 cassette into exon 2 of *cx43.4* with 24 and 48 bp homology arms resulted in 38–56% and 29–47% of injected embryos showing broad RFP expression throughout the nervous system and vasculature (*Figure 3d,f*). Increasing the length of the 5' and 3' homology arms to 1 kb did not significantly change the frequency of RFP expression compared to 24 bp (p=0.1693) or 48 bp (p=0.6520) (*Figure 3f*), with 26–47% of injected embryos showing the expected neuronal and vascular RFP expression pattern (*Supplementary file 1* Table S2-S3). Injection without the UgRNA leaves the 1 kb homology circular donor intact, and reduced targeting to 0–3% (*Figure 3f* Circular HR 1 kb; p=0.0067; *Supplementary file 1* Table S2-S3), as expected given the low frequency of homologous recombination in zebrafish embryos. For comparison with previous reports using in vitro liberation of long homology arm cassettes before injection, the 1 kb homology arm vector digested with restriction enzymes that cut within the homology arms, reducing the 5' arm to approximately 900 bp and the 3' homology arm to 700 bp. The linear DNA template was gel purified before injection. The frequency of RFP expressing embryos after injection of the linear 1 kb homology arm template was significantly reduced to approximately 5% (*Figure 3f* Linear HR 1 kb; p=0.0111; *Supplementary file 1* Table S2-S3). No expression was observed when the linear template was injected without genomic gRNA (Linear control) (*Supplementary file 1* Table S3). Together, these results suggest long homology arms do not enhance the frequency of

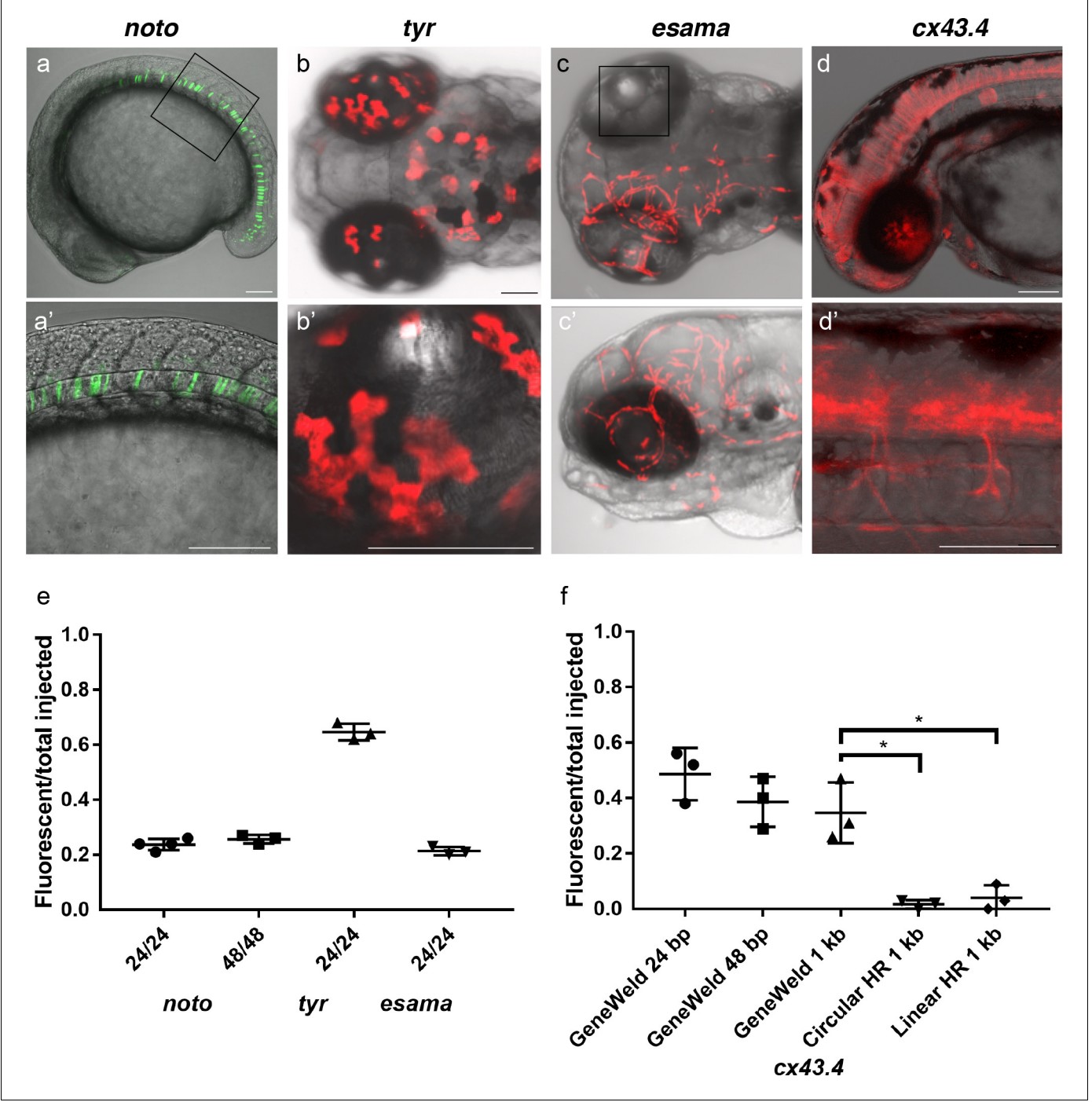

**Figure 3.** HMEJ strategy promotes efficient somatic targeting of knock-in cassettes in zebrafish. (a–d) Live confocal images of F0 injected embryos showing fluorescent reporter expression after GeneWeld targeted integration. (a, a') Mid somite stage embryo targeted at *noto* with *2A-eGFP*. (b, b') 5 days post fertilization (dpf) *Tg(UAS:mRFP)*^tpl2 embryo targeted at *tyr* with 2A-Gal4/VP16. (c) 2 dpf and (c') 3dpf *Tg(UAS:mRFP)*^tpl2 embryo targeted at *esama* with −2A-Gal4/VP16. (d, d') 31 hr post fertilization embryo targeted at *cx43.4* with 2A-tagRFP-CAAX. (e) Fraction of embryos with reporter gene expression following GeneWeld targeting at *noto*, *tyr* and *esama*. 5' and 3' homology lengths flanking donor cargos indicated in base pairs as 24/24 or 48/48. (f) Comparison of the fraction of RFP expressing embryos after targeting *cx43.4* exon 2 using GeneWeld 24/24 bp homology, GeneWeld 48/48 bp homology, Geneweld 1 kb/1 kb homology, Circular HR 1 kb/1 kb (injection did not include UgRNA, *p=0.0067), Linear HR 1 kb/1 kb (donor was digested and the linear DNA fragment containing the homology arm targeting construct was gel purified before injection, *p=0.0111). Data represents mean +/- s.e.m. of 3 independent targeting experiments. *p* values calculated using Students *t* test. Scale bars, 100 μm.

The online version of this article includes the following figure supplement(s) for figure 3:

*Figure 3 continued on next page*

*Figure 3 continued*

**Figure supplement 1.** Integration of a 2A-tagRFP reporter gene into *tyr*.

**Figure supplement 2.** Comparison of targeted integration efficiency at *esama* using short vs long homology arms and GeneWeld vs. in vitro linearized donor template.

integration at the genomic target site, compared with short 24 or 48 bp homology arms, when using the UgRNA to target double strand breaks and liberate the donor homology arms in vivo.

The comparison of short and long homology arm length on integration efficiency was also tested by targeting the 2A-Gal4/VP16 cassette to the *esama* exon two target site. Increasing the homology arms from 24 bp to 1 kb dramatically increased the percentage of RFP positive embryos, from 20–23% to 82–94% (*p*=0.0001 *Figure 3—figure supplement 2*, *Supplementary file 1* Table S3). However, the majority of RFP was not vascular specific, suggesting off-target integration of the Gal4/VP16 cassette. A high frequency of RFP-positive embryos was also observed when the donor template was injected without UgRNA (27–53%) (*Figure 3—figure supplement 2* Circular HR 1 kb) or the donor template was digested in vitro and the linear template gel purified before injection (83–94%) (*Figure 3—figure supplement 2* Linear HR 1 kb). Common repetitive elements, enhancers, or a cryptic promoter in the intronic sequence of the *esama* 1 kb homology arms may lead to off target integration and ectopic RFP expression. These results underscore the utility of short homology arms, which simplifies donor vector construction and leads to efficient precision targeted integration.

## Efficient germline transmission of precision targeted integration events

To determine the efficiency of recovering precision integration alleles through the germline, we targeted short homology arm Gal4/VP16 or tagRFP GTag cassettes into 9 independent zebrafish loci. Embryos from the *noto*, *tyr*, *esama,* and *cx43.4* GTag targeting experiments described above were raised to adulthood and outcrossed to screen for germline transmission of the reporter integrations. Three out of five (60%) adults from the *noto* injected embryos that had shown widespread notochord RFP expression transmitted a *noto-2A-TagRFP-CAAX* tagged allele through the germline (*Figure 4*, *Table 1*, *Supplementary file 1* Table S4-S5). At the *tyr* locus, although RFP expression in *tyr*-2A-Gal4/VP16 injected *Tg(UAS:mRFP)*[tpl2] embryos was highly mosaic, three out of eight (37.5%) embryos raised to adulthood transmitted germline tagged alleles (*Figure 4*, *Table 1*, *Supplementary file 1* Table S4-S5). For *esama*, 12/18 (66.7%) adults that displayed widespread vasculature RFP expression as embryos transmitted *esama-2A-Gal4/VP16* alleles to their F1 progeny (*Figure 4*, *Table 1*, *Supplementary file 1* Table S4-S5). While no germline integration events were observed with *cx43.4*, other experiments with different integration cassettes showed similar germline integration frequencies (*Table 2*, *Supplementary file 1* Table S4-S5 and data not shown). We extended the germline transmission analysis to include six additional loci: *flna*, two target sites in *msna* (exon 2 and 6), *aqp1a1*, *aqp8a1*, and *anxa2a*. Overall the data reveal high rates of germline transmission, with a combined average of 49%, that ranged from 22–100% across all targeted loci (*Figure 4*, *Table 1*, *Supplementary file 1* Table S4-S5). In addition, the founders from these experiments transmitted gametes that produced expression of RFP consistent with the targeted locus at frequencies from 2% to 64% of the F1 embryos, with an average of 17.4% (*Supplementary file 1* Table S4-S5). Together, as inferred from expression of RFP, these results suggest that the 5' and 3' short homology arms liberated with the UgRNA in vivo promote targeted integration that is efficiently transmitted through the germline in zebrafish.

## Precise 5' and 3' junctions and single copy integration in germline GTag alleles

We performed Genomic Southern blot analyses and PCR junction fragment sequencing of F1 GTag alleles to determine whether the cassettes were precisely integrated at the 5' and 3' sides of the genomic target site. Southern blot analysis and sequencing of four *tyr-2A- Gal4/VP16* F1 progeny from founder F0#1 demonstrated a single copy integration of the Gal4/VP16 cassette (*Table 2* and *Figure 4—figure supplement 1*) with precise sequence at both 5' and 3' ends of the integration site (*Table 2* and *Figure 4—figure supplement 2*). Analysis of four F1 progeny from two *noto*-2A-TagRFP-CAAX-SV40 founder adults confirmed a single copy integration in *noto* exon 1 in one of the

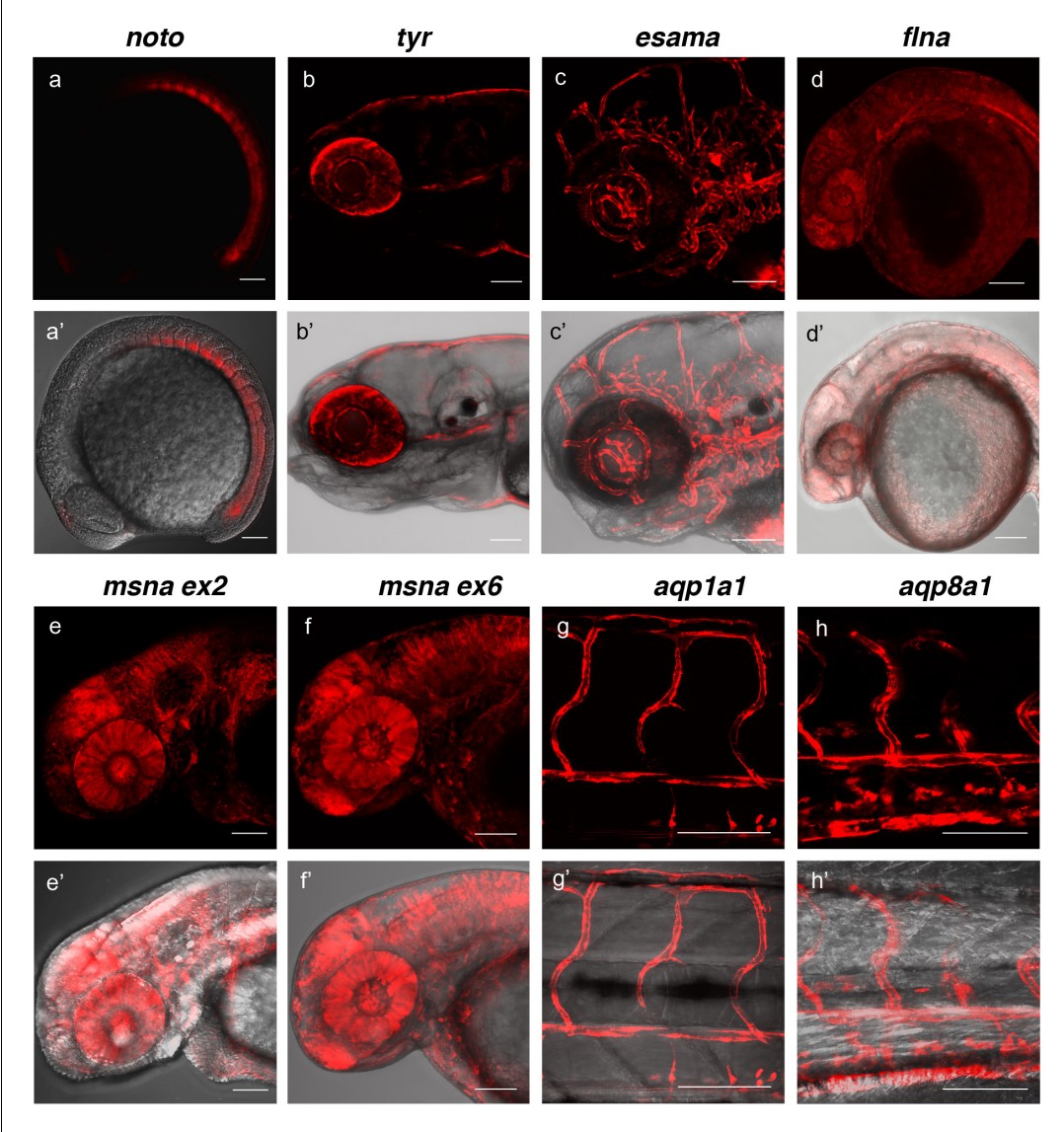

**Figure 4.** Live confocal images of F1 zebrafish with inherited germline alleles of integrated GTag reporters. (**a, a'**) *Tg(noto-2A-TagRFP)* embryo at mid somite stage showing expression in the notochord and floor plate. (**b, b'**) *Tg(tyr-2A-Gal4/VP16); Tg(UAS:mRFP)*^tpl2 5 dpf larva displaying expression in the melanocytes. (**c, c'**) *Tg(esama-2A-Gal4/VP16); Tg(UAS:mRFP)*^tpl2 4 dpf larva showing expression in the vascular system. (**d, d'**) *Tg(flna-2A-Gal4/VP16); Tg(UAS:mRFP)*^tpl2 1 dpf embryo showing widespread expression. (**e, e' and f, f'**) Exon 2 and exon 6 *msna* targeted *Tg(msna-2A-Gal4/VP16); Tg(UAS: mRFP)*^tpl2 2dpf embryos showed expression in the central nervous system and vasculature. (**g, g' and h, h'**) *Tg(aqp1a1-2A-Gal4/VP16; Tg(UAS:mRFP)*^tpl2) and *Tg(aqp8a1-2A-Gal4/VP16); Tg(UAS:mRFP)*^tpl2 2 dpf embryos display RFP expression in the trunk and tail vasculature. Scale bars, 100 μm.
The online version of this article includes the following figure supplement(s) for figure 4:

**Figure supplement 1.** Molecular analysis of F1 GeneWeld GTag targeted alleles at *tyr* and *noto*.
**Figure supplement 2.** Sequence of PCR junction fragments amplified from genomic DNA from F1 transgenic zebrafish adults generated by GeneWeld short homology directed targeted integration.
**Figure supplement 3.** Sequence of PCR junction fragments amplified from genomic DNA from F1 transgenic zebrafish adults generated by GeneWeld short homology directed targeted integration.
**Figure supplement 4.** Sequence of PCR junction fragments amplified from genomic DNA from F1 transgenic zebrafish adults generated by GeneWeld short homology directed targeted integration.

**Table 1.** Germline transmission of zebrafish GeneWeld GTag integrations.

| Genomic target | Exon | 5'/3' Homology arm length | Reporter expression | Number of germline transmitting adults | Percentage of germline transmitting adults |
|---|---|---|---|---|---|
| noto | E1 | 24/24 | 24% | 3/5 | 60% |
| tyr | E4 | 24/24 | 64% | 3/8 | 38% |
| cx43.4* | E2 | 24/24 | 50% | 0/1 | 0% |
| cx43.4* | E2 | 48/48 | 38% | 0/4 | 0% |
| esama | E4 | 24/24 | 21% | 12/18 | 67% |
| flna | E4 | 48/42 | 100% | 3/4 | 75% |
| msna | E2 | 48/48 | 55% | 1/4 | 25% |
| msna | E6 | 48/48 | 26% | 1/3 | 33% |
| aqp1a1 | E1 | 48/48 | 4% | 2/9 | 22% |
| aqp8a1 | E1 | 48/48 | 14% | 1/1 | 100% |
| anxa2a̍ | E3 | 48/48 | 35% | 4/4 | 100% |
| | | | Total | 30/61 | 49% |

F0's raised to adulthood were outcrossed and screened for germline transmission of fluorescence reporter expression. F0s transmitting/F0s outcrossed x 100 = Germline transmission percentage. At least 75 F1 embryos from each F0 adult were screened for fluorescence.

*Other experiments showed cx43.4 alleles could be transmitted through the germline in 3/11 F0 fish (27%) with a similar vector (data not shown). cx43.4 indel alleles result in sex determination defects, suggesting germline defects could contribute to variable frequencies for germline transmission of targeted integration alleles (data not shown).

Transmission is based on expression in the vasculature only.

lines (noto F0 #1 F1s), however the RFLP pattern indicated imprecise integration, while the second noto founder (noto F0#2) transmitted an allele with two RFLP fragments, suggesting an additional off-target integration (**Figure 4—figure supplement 1d-f**). Sequencing of PCR junction fragments in the noto F1 progeny revealed precise 5' integration but imprecise 3' integration, that could represent repair by NHEJ and are consistent with the Southern blot analyses (**Table 2** and **Figure 4—figure supplement 2**). While these data show recovery of a precise single copy integration at tyr, imprecise and possible off-target integrations such as those detected at noto can occur, as we previously observed when integrating a genomic reporter into noto (**Wierson et al., 2019a**).

**Table 2.** Summary of zebrafish GeneWeld GTag integrations.

| Genomic target | # of germline transmitting adults | # of precise 5' junctions by PCR | # of precise 3' junctions by PCR | # of precise integrations by Southern |
|---|---|---|---|---|
| noto | 3/5 | 8/8 | 0/3 and n/d | 0/2 |
| tyr | 3/8 | 1/1 | 1/1 | 1/1 |
| esama | 12/18 | 8/10* | 9/10* | n/d |
| flna | 3/4 | 4/4 | n/d | n/d |
| msna | 1/4 | 12/12 | 1/12 | n/d |
| aqp1a1 | 2/9 | 1/1 | 1/1 | n/d |
| aqp8a1 | 1/1 | 1/1 | 1/1 | n/d |
| Total | 25/49 (51%) | 35/37 (95%) | 13/28 (46%) | 1/3 (33%) |

F1 or F2 embryos were analyzed for junction fragments.

*Embryos from a single esama F0 founder inherited a mix of precise and imprecise junction alleles. Multiple positive FI embryos were obtained in which at least one of the embryos contained precise junctions. A polymorphism in the homology domain was also detected in the esama 5' junction from F0 #4. One of the F1s from F0#5 also contained an imprecise junction at the 5' end. esama F1 3' junctions all contain a single nucleotide variant in the homology arm. Interestingly, this was corrected to the genomic sequence. One esama F1 3' junction also included a 20 bp insertion.

n/d – not determined.

Junction fragment analysis of F1 alleles from 5 additional targeted sites in *esama*, *flna*, *msna*, *aqp1a1*, and *aqp8a1* revealed precise integration events at the 5' side for nearly all genes examined (35/37 or 95% across seven genes) (*Table 2* and *Figure 4—figure supplement 3* and *4*). This result is expected, since screening for fluorescent reporter expression selects for in-frame integration of the cassette at the 5' end. At the *esama* locus the 3' junctions were also precise in 9/10 F1s examined from 6 different F0s, and both *aqp1a1* and *app8a1* F1 alleles had precise 3' junctions (*Table 2* and *Figure 4—figure supplement 3*). Junction fragment analysis of 12 F1 *msna-2A-Gal4/VP16* from a single founder had precise 5' integrations, but only one out of the 12 had a precise 3' junction (*Figure 4—figure supplement 3*). A lower frequency of precise 3' integrations (13/28 or 46% across six genes) was observed over all loci (*Table 2* and *Figure 4—figure supplement 2–4*). Together, these results indicate that using short homology arm vectors and in vivo UgRNA liberation can promote precise, single copy integration by HMEJ at a genomic sgRNA site at high frequency in the germline, without insertion of donor vector backbone sequences, however, precision at the 3' end is reduced likely due to a lack of selection.

## Homology engineered to distant genomic sgRNA sites seeds deletion tagging in somatic tissue

To further demonstrate the utility of short homology arm directed targeted integration, we tested whether the pGTag donor could function to bridge two sgRNA genomic cuts, resulting in simultaneous deletion of intervening sequences and integration of exogenous DNA to create a 'deletion tagged' allele. Guide RNAs that target sites in the *retinoblastoma1* (*rb1*) gene were designed to exons 2 and 4, which are located 394 bp apart, and tested for efficiency (68% and 32% indel formation, *Supplementary file 1* Table S1). A more distal guide was designed in exon 25 which sits ~48.4 kb from the exon 2 sgRNA site and had 19% indel induction (*Figure 5a* and *Supplementary file 1* Table S1). The pGTag-2A-Gal4/VP16 donor contained a 48 bp 5' homology arm with sequence upstream of the sgRNA site in exon 2, and a 48 bp 3' homology arm of sequence downstream of the sgRNA site in either exon 4 or exon 25. Injection of the exon 2–4 or exon 2–25 pGTag-2A-Gal4/VP16 donor into Tg(*UAS:mRFP*)^tpl2 embryos resulted in embryos showing broad and ubiquitous RFP expression (*Figure 5b–c*). Targeting *msna* at exons 2 and 6 (88% and 98% indel formation, *Supplementary file 1* Table S1), located 7.8 kb apart, with a pGTag-2A-Gal4/VP16 donor containing 48 bp 5' exon 2 and 3' exon 6 homology arms (*Figure 5d*) resulted in RFP expression in a pattern consistent with the expression of *msna* (*Figure 5e,e'*). The frequency of RFP positive embryos was similar after targeting *rb1* exon 2–4 (44–78%) and *msna* exon 2–6 (50–85%) and did not seem to be affected by increasing the size of the deleted region from 394 bp to 48.4 Kb in *rb1* exon 2–25 (49–70%) (*Figure 5f*). Somatic junction fragment analysis detected precise integration of the 2A-Gal4/VP16 cassette in both genes at the 5' upstream exon (*rb1* 97%; *msna* 85%) and 3' downstream exon (*rb1* 67%; *msna* 45%) (*Figure 5—figure supplement 1*). However, only one out of 16 (6%) *rb1* targeted F0 founders transmitted a *rb1-e2-25-2A-Gal4/VP16* integration allele through the germline (*Supplementary file 1* Table S4-S5). The allele contained a precise 5' junction at the exon 2 target site, but the 3' junction could not be amplified by PCR. None of the 10 *msna* e2-e6 2A-Gal4/VP16 targeted F0 zebrafish transmitted a deletion tagged allele to the next generation (*Supplementary file 1* Table S4-S5). In contrast, targeting 2A-Gal4/VP16 to *msna* exon 2 or 6 alone resulted in 2 out of 7 F0s transmitting a targeted allele to the next generation (*Supplementary file 1* Table S4-S5).

Together, these results demonstrate simultaneous targeting of two distal genomic cut sites in somatic tissue can create precise integration at both ends of a pGTag reporter cassette, but these events were not efficiently passed through the germline. We attempted deletion tagging at three additional loci, *kdrl*, *s1pr1*, and *vegfaa*, which showed 32–81% expression in F0 embryos, but did not recover germline transmission to the F1 generation (*Supplementary file 1* Table S2-S5). While HMEJ driven by short homology arms and liberation by the UgRNA can efficiently promote precision targeted integration at a single double strand break in somatic and germline tissue, efficient simultaneous deletion and integration to bride two target sites in the germline likely occurs at a much lower frequency.

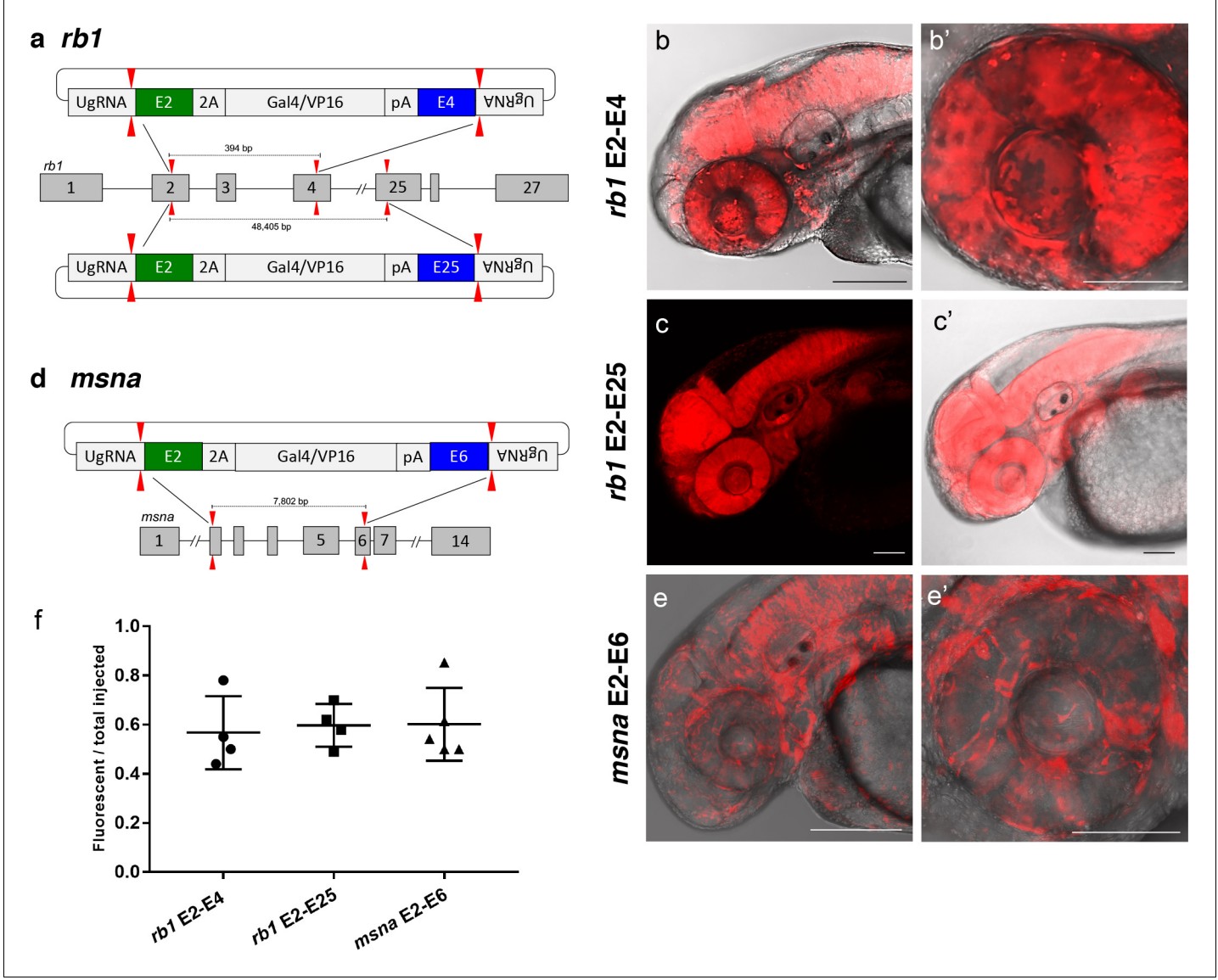

**Figure 5.** Deletion tagged alleles created with the GeneWeld strategy in zebrafish somatic tissue. (**a**) Schematic for Gal4/VP16 reporter integration to tag a deletion allele of *rb1* exons 2–4 (top) and *rb1* exons 2–25 (bottom). Arrowheads designate CRISPR/Cas9 DSBs. CRISPR sgRNAs in two exons are expected to excise the intervening genomic DNA. The targeting vector contains a 5' homology arm flanking the upstream exon target site and a 3' homology arm flanking the downstream exon target site. (**b, b'**) Live confocal image of F0 Tg(*UAS:mRFP*)[tpl2] embryo after 2A-Gal4/VP16 deletion tagging at *rb1* exons 2–4. (**c, c'**) Live confocal image of F1 *Tg(rb1-e2-2A-Gal4/VP16)* embryo from a founder targeted at *rb1* exons 2–25. A deletion from exon 2–25 was not observed in the F1 generation, but the 5' junction was in frame. (**d**) Schematic for 2A-Gal4/VP16 deletion tagging of *msna* exons 2–6. (**e, e'**) Live confocal image of F0 Tg(*UAS:mRFP*)[tpl2] embryo after 2A-Gal4/VP16 deletion tagging at *msna* exons 2–6. (**f**) Somatic reporter efficiency of targeted deletion tagging using 48 bp homology arms for *rb1* exons 2–4, *rb1* exons 2–25, and *msna* exons 2–6. Data represents mean +/- s.e.m. of 4 (*rb1*) and 5 (*msna*) independent targeting experiments. Scale bars 200 μm (**b, c, c', e**); 100 μm (**b', e'**).

The online version of this article includes the following figure supplement(s) for figure 5:

**Figure supplement 1.** Sequences of 5' and 3' junction fragments from *rb1* exon 2–4, *rb1* exon 2–25, and *msna* exon 2–6 deletion tagged alleles in F0 injected embryos.

## Integration of exogenous DNA using HMEJ in porcine and human cells is more efficient than HR

To determine if HMEJ integration directed by short homology functions efficiently in large animal systems, we tested our targeting strategy in *S. scrofa* fibroblasts (*Figure 6a–c*). A cassette that drives ubiquitous eGFP expression from the UbC promoter (*Figure 6a*) was designed based on the pGTag

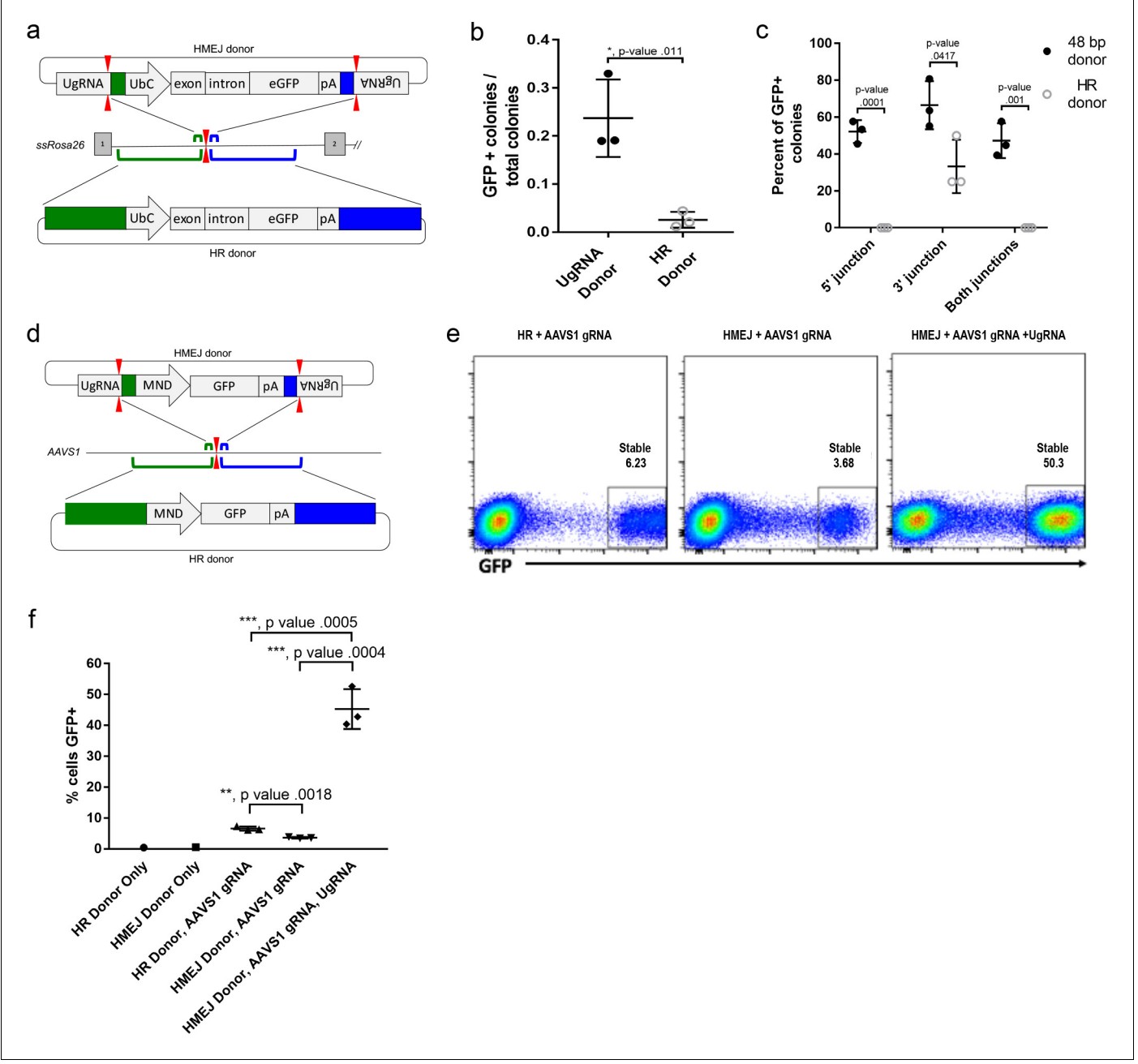

**Figure 6.** HMEJ-based targeted integration with UgRNA-based vectors promotes efficient knock-in in porcine fibroblasts and human K-562 cells. (**a**) Strategy for integration using HMEJ and HR donors into intron 1 of *S. scrofa ROSA26* locus. Arrowheads CRISPR/Cas9 (for HMEJ donor) and TALEN (genome) DSBs. (**b**) Targeting efficiency of the HMEJ donor vs the HR donor as reported by GFP positive colonies out of total colonies. (**c**) Percent of GFP positive colonies analyzed containing properly sized junction fragments, comparing HMEJ and HR donors. Data are from three independently targeted cell populations. Data represents mean +/- s.e.m. of 3 independent targeting experiments. (**d**) Diagram of HR and HMEJ strategies for targeted integration of a MND:GFP reporter cassette into the human *AAVS1* locus. (**e**) Flow cytometry analysis of GFP expression 14 days post-electroporation for each targeting modality: HR (left), HMEJ without universal sgRNA (middle), and HMEJ with universal sgRNA (right). Stable gate was drawn to measure the uniformly expressing population formed by targeted integration and was set based on episome only controls. (**f**) Quantification of stable GFP expressing population as measured by flow cytometry at day 14. Data are from three independently targeted cell populations. Data represents mean +/- s.e.m. of 3 independent targeting experiments. p values calculated using two-tailed unpaired t-test.

The online version of this article includes the following figure supplement(s) for figure 6:

**Figure supplement 1.** HMEJ-mediated targeted integration of an MND:GFP reporter at the *AAVS1* locus in human K-562 cells.

**Figure supplement 2.** Direct sequencing of 5' junction PCR products derived from three independently targeted bulk cell populations.

vector series, with UgRNA sites on either side of 48 bp homology arms that flank the cassette (UgRNA donor). The 5' and 3' homology arms contained sequences that map to a TALEN cut site in intron 1 of the *ROSA26* safe harbor locus. Fibroblasts were electroporated with TALENs, UgRNA, and Cas9 RNAs, and the *ROSA26* UbC:GFP UgRNA targeting vector. The efficiency of precision integration was compared to cells electroporated with just the TALEN pair and a donor containing 760 bp homology arms (HR Donor), which would integrate through homologous recombination (HR). GFP expression was observed in 23% of colonies using the UgRNA donor, compared to 2% of colonies using the HR donor with 760 bp homology arms (*Figure 6b*). 5' and 3' junctions were observed in over 50% of the UgRNA donor GFP+ colonies while none of the HR donor GFP+ colonies contained both junctions (*Figure 6c*). Sequencing of junctions from 8 UgRNA donor GFP+ colonies that were positive for both junctions showed precise integration in 7/8 colonies at the 5' junction and 8/8 colonies at the 3' junction.

We next tested the efficiency of HMEJ targeted integration in human K-562 cells using a donor vector with a MND:GFP reporter (*Halene et al., 1999*) and 48 bp homology arms designed to target a CRISPR/Cas9 site in the *AAVS1* safe harbor (*Figure 6d–f*). K-562 cells were electroporated with *AAVS1* sgRNA, UgRNA, Cas9 and the UgRNA HMEJ donor or an HR donor with long 806 bp homology arms. Cells were FACs sorted by GFP at day 14 following electroporation. Over 50% of cells targeted with the HMEJ 48 bp homology arm donor were GFP positive, compared to only 6% of cells electroporated with the 806 bp homology arm HR donor, indicating HMEJ promoted efficient integration and stable expression of the MND:GFP cassette at the *AAVS1* locus (*Figure 6—figure supplement 1*). GFP expression was maintained over 50 days, and 5' precise junction fragments were observed following PCR amplification in bulk cell populations (*Figure 6—figure supplement 2*). These results demonstrate that the HMEJ strategy using short homology arms outperforms traditional HR techniques for targeted integration in mammalian cell systems and is effective without antibiotic selection.

## Discussion

Here we adapted the short homology arm targeted integration approach described by *Hisano et al. (2015)* to develop a vector suite, pGTag, and detailed protocol for efficient site-directed gene targeting in zebrafish. We extended this strategy to mammalian cells and show efficient recovery of precision targeted integration at safe harbor loci in pig fibroblasts and human K-562 cells. The pGTag vector series has been deposited at Addgene, and the open source protocol and homology arm website design tool are available at The Gene Sculpt suite (genescultp.org) (*Mann et al., 2019*). We engineered our targeting vectors with knock-in cassettes flanked by enzyme sites to clone short homology arms that are first assembled by oligonucleotide annealing. We also placed a universal sgRNA sequence outside the homology arm cloning sites, which allows for Cas9 liberation of the targeting cassette in vivo. The pGTag cassettes contain 2A-fluorescent protein or 2A-Gal4/VP16 reporters for visualizing integrated reporter expression in injected embryos. Together, these design elements, that include simultaneous double strand breaks by Cas9 at the genomic target site and vector universal sgRNA sites, and preselection of fluorescent reporter expression that requires in frame fusion and precise 5' targeted integration, may enhance the efficient recovery of germline alleles over previous reports in zebrafish (*Hisano et al., 2015*; *Hoshijima et al., 2016*; *Shin et al., 2014*). Our results show 49% of all adult zebrafish preselected for on target reporter expression as embryos transmitted precision targeted knock-in alleles through the germline, with 17.4% of gametes carrying the edited allele of interest (*Supplementary file 1* Table S4-S5). We demonstrated efficient targeted integration of cargos up to 2 kb in length in zebrafish, pig fibroblasts, and human K-562cells. Both CRISPR/Cas9 and TALEN genomic sites are targeted with high efficiency, providing flexibility in design and increasing genome-wide accessibility.

Our initial targeting experiments at *noto* using a single 5' homology arm and single *noto* sgRNA to target the genome and donor vector showed an increase in targeting efficiency as the length of the homology arm increased from 12 to 48 bp. However, when targeting was performed with homology on both sides of the cargo flanked by the universal UgRNA sites, short and long homology arms were equally efficient at targeted integration (*Figure 3e*). This may reflect enhanced efficiency of in vivo cargo liberation by the UgRNA. Positive integration events of tagged alleles are selected by fluorescence reporter expression, reflecting the endogenous pattern of expression of

the targeted gene. Our observations indicate this correlates with predominantly precise integration events as analyzed by PCR and sequencing in both somatic tissue and through the germline. We speculate that inclusion of homology at the 3' end of our cargo creates competition for the donor DNA ends, as not all editing events are precise at both 5' and 3' junctions (*Figure 4—figure supplement 1* and *2*). Thus, it is conceivable that precise events at the 3' end could preclude precise integration at the 5' end during some editing events, and vice versa. It is tempting to speculate that these data hint at synthesis dependent strand annealing (SDSA) as a possible DNA repair mechanism for pGTag donor integration (*Ceccaldi et al., 2016*). After strand invasion using either of the homology domains and replication through the reporter, second DNA end capture may abort before or after replication through the opposing homology domain, resulting in imprecision, as greater than or equal to 150 bp is required for proper second end capture in yeast (*Mehta et al., 2017*). Our targeting results are similar to a recent study in zebrafish that reported short homology directed integration of a 2A fluorescent reporter near the termination codon at three genes (*Luo et al., 2018*). Our study robustly demonstrates the short homology targeted integration strategy can be reproducibly applied at many loci to generate zebrafish knock-in mutations which transmit through the germline at high frequency.

Using two sgRNAs we attempted to simultaneously delete a section of the *rb1* gene and introduce a 2A-Gal4/VP16 reporter cassette to create a deletion tagged allele. While we found evidence of efficient deletion and tagging in F0 targeted embryos, we were unable to recover germline alleles. It is possible that the inability to recover *rb1* tagged alleles was due to the essential requirement for this tumor suppressor gene, in either somatic tissue or germline stem cells. Our previous targeted mutagenesis at *rb1* revealed it is necessary to significantly reduce the amount of TALENs or CRISPR/Cas9 sgRNA injected into the embryo in order to prevent complete bi-allelic inactivation and allow for recovery of viable mosaic adults that transmit indel mutations through the germline (*Solin et al., 2015*). Combined with previous work that indicates stem cells are susceptible to apoptosis following gene editing (*Ihry et al., 2018*; *Li et al., 2018*), these factors may have contributed to the lack of recovery of *rb1* deletion tagged alleles. Additional experiments at non-essential genes are needed to determine the efficiency of simultaneous gene deletion and tagging.

Previous studies in zebrafish examined whether homology arm length (*Shin et al., 2014*) or in vivo I-SceI cleaved, linear templates with exposed homology arms (*Hoshijima et al., 2016*) would enhance targeting efficiency. We compared the effect of short and long homology arm length on targeted integration in somatic tissue using our pGTag vectors and targeting strategy. Increasing the *cx43.4* 5' and 3' homology arms to 1 kb did not result in a significant difference in targeting efficiency compared to vectors with short 24 bp or 48 bp homology arms, as assayed by fluorescence (*Figure 3*). As expected, injection of the long homology arms vector without the universal sgRNA to liberate the arms effectively eliminated reporter expression and targeted integration. We also found there may be a significant disadvantage to long homology arms. Injection of the *esama* 2A-Gal4/VP16 vector with 1 kb homology arms led to nearly 100% of embryos showing widespread RFP expression, and 30–60% of embryos injected with circular plasmid alone were reporter positive (*Figure 3—figure supplement 2*). Together, these results suggest long homology arms of intronic or intergenic sequences may contain repetitive elements that drive off-target integration across the genome. Alternatively, intronic promoters or enhancers could be unmasked by integration at ectopic sites, leading to high rates of false positives. Given the simple design, low cost, and ease of assembly of short homology arms, this approach may be preferable for most precision targeted integration experiments. However, further experiments are required at additional loci to determine whether the advantages of using short homology in targeting vectors represents a universal rule.

Our results show short homology arms are effective at directing precision targeted integration at CRISPR/Cas and TALEN sites in mammalian cells without selection by antibiotic resistance. Introducing a universal targeting vector with 48 bp homology arms into pig fibroblasts or human K562 cells lead to a 10-fold increase in targeted integration in comparison to a circular template that would integrate by homologous recombination. This is in contrast to previous studies in mammalian cells that indicated the frequency of targeted integration was not significantly enhanced until the length of homology arms was increased to 600 bp (*Zhang et al., 2017*). This discrepancy may reflect variation in experimental design or activity of DNA repair pathways in different cell types. Our results showing robust precision targeted integration using short homology arms in zebrafish and two mammalian cells lines suggest this simple, straightforward approach will be broadly applicable across

species and model systems. In support of this, a recent study has shown 100 bp short homology arms and flanking CRISPR sgRNA sites efficiently directs targeted integration of DNA cassettes in *Drosophila* and S2 cells (*Kanca et al., 2019*).

In summary, we have shown short homology arm directed targeted integration is highly efficient in zebrafish and mammalian cell lines. The simplicity of our strategy for building arms into the pGTag vectors, a universal guide to liberate the targeting cassette, and the GeneSculpt Suite website together will aid in streamlining targeted integration experimental design. Our vector suite provides a platform that can expand the zebrafish functional genomics toolkit for efficient site-directed modifications that create gene fusions, introduce cDNA variants, recombinases, or floxed gene-breaking cassettes (*Clark et al., 2011*) for conditional gene inactivation. Targeted integration will allow tagged or mutant genes to be expressed at endogenous levels and avoid ectopic or over-expression that can result from random transgene integration. These tools will broaden the use of designer nucleases for homology-based gene editing at CRISPR/Cas9 and TALEN cut sites in zebrafish and mammalian cell lines. Our results open the door for more advanced genome edits in animal agriculture and human cell lines.

# Materials and methods

## Key resources table

| Reagent type (species) or resource | Designation | Source or reference | Identifiers | Additional information |
|---|---|---|---|---|
| Gene (*Danio rerio*) | *anxa2a* | | ensemble: ENSDAR G00000003216 | |
| Gene (*Danio rerio*) | *aqp1a1* | | ensemble: ENSDAR G00000023713 | |
| Gene (*Danio rerio*) | *aqp8a1* | | ensemble: ENSDAR G00000045141 | |
| Gene (*Danio rerio*) | *cx43.4* | | ensemble: ENSDAR G00000007099 | |
| Gene (*Danio rerio*) | *esama* | | ensemble: ENSDAR G00000077039 | |
| Gene (*Danio rerio*) | *flna* | | ensemble: ENSDAR G00000074201 | |
| Gene (*Danio rerio*) | *kdrl* | | ensemble: ENSDAR G00000105215 | |
| Gene (*Danio rerio*) | *mmp14a* | | ensemble: ENSDAR G00000002235 | |
| Gene (*Danio rerio*) | *msna* | | ensemble: ENSDAR G00000058128 | |
| Gene (*Danio rerio*) | *rb1* | | ensemble: ENSDAR G00000006782 | |
| Gene (*Danio rerio*) | *s1pr1* | | ensemble: ENSDAR G00000042690 | |
| Gene (*Danio rerio*) | *tyr* | | ensemble: ENSDAR G00000039077 | |
| Gene (*Danio rerio*) | *vegfaa* | | ensemble: ENSDAR G00000103542 | |
| Gene (*Homo sapiens*) | *AAVS1* | | HGCN:22 Adeno-Associated Virus Integration Site 1 | (*Kotin et al., 1992*) |
| Gene (*Sus scrofa*) | *ROSA26* | This paper | Porcine homolog of mouse *ROSA26* safe harbor locus for transgene integration | |

*Continued on next page*

Continued

| Reagent type (species) or resource | Designation | Source or reference | Identifiers | Additional information |
|---|---|---|---|---|
| Strain, strain background (*Escherichia coli*) | NEB Stable Competent *E. coli* | New England Biolabs | C3040I | Electrocompetent Cells |
| Strain, strain background (*Escherichia coli*) | One Shot TOP10 Chemically Competent *E. coli* | Thermo Fisher/Invitrogen | C404010 | Electrocompetent Cells |
| Genetic reagent (*Danio rerio*) | WIK | Zebrafish International Resource Center | ZIRC:ZL84 | Wildtype strain of zebrafish |
| Genetic reagent (*Danio rerio*) | Tg(*UAS:mRFP*)tpl2 | Balciunas lab | Tg(*miniTol2 < 14XUAS:mRFP, γCry:GFP>*)tpl2 | Maintained in the lab of D. Balciunas (*Balciuniene et al., 2013*) |
| Cell line (*Homo sapiens*) | K562 | ATCC | ATCC:CCL-243 | chronic myelogenous leukemia cell line |
| Cell line (*Sus scrofa*) | Porcine fibroblast cell line | This paper | | Recombinetics, Inc |
| Transfected construct (*Homo sapiens*) | p*AAVS1*-MND:GFP | This paper | | B. Moriarity lab |
| Transfected construct (*Sus scrofa*) | p*ROSA26* UbC:GFP UgRNA | This paper | | Recombinetics, Inc |
| Recombinant DNA reagent | pT3TS-nCas9n | Wenbiao Chen | Addgene:46757 | Plasmid for in vitro synthesis of Cas9 mRNA |
| Recombinant DNA reagent | p494-2a-TagRFP-CAAX-SV40 | This paper | | available from J. Essner lab; Deposited at Addgene |
| Recombinant DNA reagent | pGTag-2A-TagRFP-CAAX-SV40 | This paper | | available from J. Essner lab; Deposited at Addgene |
| Recombinant DNA reagent | pGTag-2A-Gal4/VP16-βactin | This paper | | available from J. Essner lab; Deposited at Addgene |
| Recombinant DNA reagent | pGTag-2A-eGFP-SV40 | This paper | | available from J. Essner lab; Deposited at Addgene |
| Sequence-based reagent | | This paper | PCR primers and oligos | See *Supplementary file 1* |
| Commercial assay or kit | pCR4 TOPO TA Cloning Kit | ThermoFisher/Invitrogen | ThermoFisher:K457502 | |
| Commercial assay or kit | Zero Blunt TOPO PCR Cloning Kit | ThermoFisher/Invitrogen | ThermoFisher:K2800J10 | |
| Commercial assay or kit | NEBNext Ultra II DNA Library Prep Kit for Illumina | New England Biolabs | NEB:E7645L | For MiSeq multiplex DNA sequencing |
| Software, algorithm | CRISPRScan | A. Giraldez lab | http://www.crisprscan.org/ | (*Moreno-Mateos et al., 2015*) |

*Continued on next page*

*Continued*

| Reagent type (species) or resource | Designation | Source or reference | Identifiers | Additional information |
|---|---|---|---|---|
| Software, algorithm | pGTag | This paper | http://genesculpt.org/gtaghd/ and https://github.com/Dobbs-Lab/GTagHD *Wierson et al., 2019b* copy archived at https://github.com/elifesciences-publications/GTagHD | short homology arm design |
| Software, algorithm | ICE | Synthego | Inference of CRISPR Edits (ICE) https://www.synthego.com/products/bioinformatics/crispr-analysis | Indel analysis of Sanger sequenced DNA |
| Software, algorithm | Cas-Analyzer | RGEN | CRISPR RGEN Tools http://www.rgenome.net/cas-analyzer/#! | Indel analysis of NextGen sequenced DNA |

## Contact for reagent and resource sharing

Further information and requests for resources and reagents should be directed to Jeffrey Essner (jessner@iastate.edu).

## Zebrafish strains and mammalian cell lines

Zebrafish were maintained on an Aquatic Habitats (Pentair) aquaculture system at 27℃ on a 14 hr light/10 hr dark cycle. Wild-type WIK were obtained from the Zebrafish International Resource Center (https://zebrafish.org/home/guide.php). The transgenic line Tg(*miniTol2 <14XUAS:mRFP, γCry: GFP>*)$^{tpl2}$, referred to as Tg(*UAS:mRFP*)$^{tpl2}$, was previously described (*Balciuniene et al., 2013*). All zebrafish experiments were carried out under approved protocols from Iowa State University Animal Care and Use Committee Log#11-06-6252, in compliance with American Veterinary Medical Association and NIH guidelines for the humane use of animals in research.

The human K-562 chronic myelogenous leukemia cell line (ATCC CCL-243) used in gene targeting experiments was cultured at 37℃ in 5% $CO_2$ in RPMI-1640 medium (Thermo Fisher Scientific) supplemented with 10% fetal bovine serum (FBS) and Penicillin/Streptomycin. Electroporation was conducted with $1.5 \times 10^5$ cells in a 10 µl tip using the Neon electroporation device (Thermo Fisher Scientific) with the following conditions: 1450V, 10 ms, 3x pulse. Nucleic acid dosages were as follows: 1.5 µg Cas9 mRNA (Trilink Biotechnologies), 1 µg each chemically modified sgRNA (Synthego), and 1 µg donor plasmid.

Porcine fibroblasts were cultured in DMEM (high glucose) supplemented to 10% vol/vol FBS, 20 mM L-glutamine and 1X Pen/Strep solution and transfected using the Neon system (Invitrogen). Briefly, $1 \times 10^6$ fibroblasts were transfected with 1 ug of polyadenylated ROSA TALEN mRNA, 1 µg of universal UgRNA, 1 µg of polyadenylated Cas9 mRNA and 1 µg of donor plasmid. Transfected cells were cultured for 3 days at 30℃ before low density plating, extended culture (10 days) and colony isolation. Individual colonies were aspirated under gentle trypsanization, replated into 96- well plates and cultured for 3–4 days.

## pGTag series vectors

To build the pGTag vector series, 2A-TagRFP, 2A-eGFP, and 2A-Gal4/VP16 cassettes were assembled from a 2A-TagRFP-CAAX construct, p494. To clone the eGFP cassette, the plasmid p494 was amplified with primers F-p494-XhoI and R-p494-SpeI to generate unique enzyme sites in the backbone. The eGFP coding sequence (Clontech Inc) was amplified with the primers F-eGFP-SpeI and R-eGFP-XhoI to generate the corresponding enzyme sites on the eGFP coding sequence. Fragments

were digested with SpeI-HF and XhoI (NEB) and following column purification with the Qiagen mini-prep protocol, were ligated to the plasmid backbone with T4 ligase (Fisher).

The Gal4/VP16 coding sequence and zebrafish β-actin 3' untranslated region was amplified from vector pDB783 (*Balciuniene et al., 2013*) with primers F-2A-Gal4-BamHI and R-Gal4-NcoI to add a 2A peptide to the 5' end of the Gav4Vp16 cDNA. The resulting PCR product was then cloned into the intermediate Topo Zero Blunt vector (Invitrogen) and used for mutagenesis PCR with primers F and R '-gal4-Ecofix' to disrupt the internal EcoRI restriction site. The resulting Gal4/VP16 sequence was cloned into the BamHI and NcoI sites in the p494 backbone.

The 5' universal guide RNA UgRNA site and *lacZ* cassette were added to pC-2A-TagRFP-CAAX-SV40, pC-2A-eGFP-SV40, and pC-2A-Gal4/VP16-β-actin with the following steps. The *lacZ* was first amplified with primers F-lacZ and R-lacZ, which add the type IIS enzyme sites to either end of the *lacZ*. The resulting PCR product was then cloned into an intermediate vector with the Zero Blunt TOPO PCR Cloning Kit (Invitrogen). This intermediate was used as a template in a nested PCR to add the Universal guide sequence GGGAGGCGTTCGGGCCACAGCGG to the end of the *lacZ* sequence. The nested PCR used primers F-lacZ-universal-1 and R-lacZ-universal-BamHI to add the first part of the universal guide to one end and a BamHI site to the other. This was used as template for PCR with the primers F-lacZ-universal-EcoRI and R-lacZ-universal-BamHI to add the remainder of the universal guide and an EcoRI site. The fragment was column purified as above, digested with EcoRI-HF and BamHI-HF and cloned into the appropriate sites in the three vectors.

The 3' universal guide RNA UgRNA site and type 2 restriction enzyme sites were cloned into each vector in two steps. A segment from a Carp β-actin intron containing a 99 bp spacer flanked by two BspQI sites was amplified using the primers F-3'-uni-1 and R-3'-uni-1 to add the universal site to one side of the spacer. This product was column purified as above and used as template for the second amplification with primers F-3'-uniNco1 and R-3'-uniEagI to add cloning sites. This product was column purified and cloned using the Topo zero blunt kit. This intermediate was digested with NcoI-HF and EagI, and the BspQI fragment purified and cloned into the three vectors as above. Ligations were grown at 30℃ to reduce the possibility of recombination between the two universal guide sites.

Correct clones for pU-2A-TagRFP-CAAX-U, pU-2A-eGFP-U, and pU-2A-Gal4/VP16-U were selected and used as template for mutagenesis PCR with KOD to remove extra BspQI sites from the backbone with primers F/R-BBfix, digested with DpnI (NEB), and ligated with T4 ligase. A correct pU-2A-TagRFP-CAAX-U clone was used as template for PCR with F/R-TagRFPfix to interrupt the BspQI site in the TagRFP coding sequence as above. A correct clone of pU-2A-Gal4/VP16-U was selected and used as template with primers F/R-Bactfix to remove the BspQI site in the β-actin terminator, the product was re-cloned as above. All constructs were sequence verified.

## sgRNA target site selection, homology arm design and pGTag donor vector construction

CRISPR/Cas9 target sites in exons of zebrafish genes were identified using CRISPRScan (http://www.crisprscan.org/; *Moreno-Mateos et al., 2015*). 5' and 3' homology arms of specified length directly flanking a genomic targeted double strand break were generated by annealing two complementary oligonucleotides. The double stranded 5' and 3' homology arms with appropriate overhangs were cloned into the pGTag vector BfuAI and BspQI sites, respectively, flanking the cargo. A three-nucleotide buffer sequence lacking homology to the genomic target site was engineered between the donor UgRNA PAM and the 5' end of the homology arms. This was done in case the UgRNA PAM sequence was complementary to the nucleotides located 5' to the start of the homology arm, which would increase the 24 or 48 bp homology arm length. Maps for the pGTag vectors and an open source protocol for cloning the homology arms are available at http://genesculpt.org/gtaghd/. The pGTag vectors are available through Addgene (https://www.addgene.org/kits/essner-geneweld/).

To generate 1 kb homology arms for zebrafish *cx43.4* and *esama*, ~2 kb of genomic DNA surrounding the CRISPR target site was PCR amplified from adult WIK finclips using the proofreading enzyme KOD (EMD Millipore), and then sequenced to identify polymorphisms. Primers were designed to sit 1032 bp up and down stream of the cut site according to the Ensemble.org reference genome V11. Primers also contain either BfuAI and BspQI recognition sequences to make the appropriate overhangs for Golden Gate cloning into a pGTag vector or sequence for Gibson cloning into a pGTag vector. PCR was performed with the proofreading polymerase KOD and using

genomic DNA from animals homozygous for the most common polymorphisms was used as template. The products were then Topo Blunt (Thermo Fisher Scientific) cloned for sequencing. The homology arms were Golden Gate or Gibson cloned into a pGTag vector containing the same cassette as the previous injections for the target locus. pGTag vectors with 1 kb homology arm vectors were injected into embryos from adults with the matching genomic sequence. *Supplementary file 1* Table S5 lists the sequences of all homology arms, sgRNA target sites, and spacers. For each locus injections were done in triplicate, and for those targeting the locus with 1 kb homology arms the following controls were also performed; plasmid only, plasmid with universal gRNA and without genomic gRNA, and plasmid linearized in vitro with genomic gRNA.

## Zebrafish embryo injection

The pT3TS-nCas9n expression vector for in vitro synthesis of nls-Cas9-nls mRNA was a gift from Wenbiao Chen (Addgene plasmid # 46757). XbaI linearized pT3TS-nCas9n was purified under RNase-free conditions with the Promega PureYield Plasmid Miniprep System. Linear, purified pT3TS-nCas9n was used as template for in vitro transcription of capped, polyadenylated mRNA with the Ambion T3TS mMessage mMachine Kit. mRNA was purified using the Qiagen miRNeasy Kit. The genomic and universal sgRNAs were generated using cloning free sgRNA synthesis as described in *Varshney et al. (2015)* and purified using Qiagen miRNeasy Kit. Donor vector plasmid DNA was purified with the Promega PureYield Plasmid Miniprep System.

sgRNA targeted mutagenesis efficiency was determined by measuring the frequency of indel mutations at the target site. Individual embryos were injected with 150 pg Cas9 mRNA and 25 pg sgRNA. PCR amplicons over the cut site were first examined for mutagenesis (or a smear) on a 2–3% agarose gels in 1X TAE. For the *flna* exon 4 and *msna* exon 2 sgRNAs, a single 2 dpf embryo was placed in 15 ul of 50 mM NaOH, heated at 95℃ for 30 min to extract genomic DNA, and neutralized by addition of 1.5 ul of 1M Tris pH 8.0. 1 ul of DNA extract was used as template for PCR to generate amplicons for sequence analysis. The *flna* exon 4 and *msna* exon 2 amplicons were sequenced directly by Sanger sequencing at the Iowa State University DNA Facility, and indel frequency determined using the Inference of CRISPR Edits (ICE) analysis tool at Synthego (https://www.synthego.com/products/bioinformatics/crispr-analysis). For the *noto* exon 1, *cx43.4* exon 2, *esama* exon 2, *msna* exon 6, *rb1* exon 2, exon 4 and exon 25, *aqp1a1* exon 1 and *aqp8a1* exon 1 sgRNAs, at 3 dpf 5 embryos were pooled in 50 ul of 50 mM NaOH, and 2 ul used as template for PCR. Barcoded libraries were prepared using NEBNext Ultra II DNA Library Prep Kit for Illumina (NEB #E7645L) for MiSeq 250 bp single read sequencing at the University of Kansas Genome Sequencing Core. Indel analysis of MiSeq reads was performed using Cas-Analyzer at CRISPR RGEN Tools (http://www.rgenome.net/cas-analyzer/#!). MiSeq data can be found at: Essner, Jeffrey; *McGrail et al., 2011*, MiSeq data for Cas9, Dryad, Dataset, https://doi.org/10.5061/dryad.m63xsj3zc.

Linear targeting cassettes for *cx43.4* and *esama* were generated by restriction enzyme digestion with enzymes that cut in or adjacent to the 1 kb homology arms, followed by gel isolation and re-purification with the Promega PureYield Plasmid Miniprep System. The resulting linear fragments were similar in size to the circular vectors containing short homology arms. The *cx43.4* vector was digested with NdeI, which truncates the 5' homology arm to 900 bp, and EcoRI, which truncates the 3' homology arm to 700 bp. In the *esama* vector the EcoRI site sits upstream and next to the 5' universal guide RNA sequence, leaving 31 bp of non-homologous sequence on the 5' end of the 1 kb arm. The EagI site sits 16 bp downstream of the 3' UgRNA target sequence, adding 43 bp of non-homologous sequence at the 3' end of the 3' homology arm.

All genes were targeted by injection into the cytoplasm of the 1 cell stage embryo 2 nl of solution containing 150 pg of nCas9n mRNA, 25 pg of genomic sgRNA, 25 pg of UgRNA, and 10 pg of donor DNA diluted in RNAse free ddH$_2$O with the exception of *rb1*. The *rb1* targeting mixture contained 300 pg nCas9n mRNA. Gal4/VP16 pGTag donors were injected into embryos from the UAS mRFP reporter line Tg(*miniTol2 <14XUAS:mRFP, γCry:GFP>*)$^{tpl2}$ (*Balciuniene et al., 2013*).

## Zebrafish targeted integration junction fragment analysis and recovery of germline alleles

Injected embryos were screened for fluorescence reporter expression at 24, 48 and 72 hr post fertilization on a Zeiss Discovery dissection microscope. Genomic DNA for PCR was extracted by

digestion of single embryos in 50 mM NaOH at 95℃ for 30 min and neutralized by addition of 1/10[th] volume 1M Tris-HCl pH 8.0. Junction fragments were PCR-amplified with primers listed in *Supplementary file 1* Table S6 and the PCR products TOPO-TA cloned before sequencing. For live imaging of reporter expression, embryos were mounted on slides in 1.2% low-melt agarose in 160 ug/ml tricaine methanesulfonate, and images were captured on a Zeiss LSM 700 laser scanning confocal microscope. RFP or GFP positive embryos were raised to adulthood and outcrossed to wild-type WIK adults to test for germline transmission of fluorescence in F1 progeny. Adults that were injected as embryos with Gal4/VP16 constructs targeting *tyr*, *esama*, *rb1* and *msna* were crossed to Tg(*miniTol2 <14XUAS:mRFP, γCry:GFP>*)[tpl2] (*Balciuniene et al., 2013*).

## Zebrafish genomic southern blot analysis

Genomic Southern blot and knock-in cassette copy number analysis of zebrafish F1 progeny carrying targeted integration alleles was performed as described previously (*McGrail et al., 2011*). Adult zebrafish were euthanized in ice cold water, flash frozen in liquid nitrogen, and tissues ground with a mortar and pestle. Genomic DNA was extracted from ground tissue using the Qiagen Blood and Cell Culture Maxi Kit (Qiagen). 10 ug of genomic DNA was digested with restriction enzymes, electrophoresed, and blotted using a Whatman Turboblotter Rapid Downward Transfer System (ISC Bio-Express). DIG labeled probes were synthesized with PCR-DIG Probe Synthesis Kit (Roche), and hybridization and chemiluminescence detection were performed with DIG Easy Hyb Granules/Wash and Block Buffer Set and CSPD (Roche). Images were captured and analyzed on a BioRad ChemiDoc XRS system. PCR primers used for genomic and donor specific probes are listed in *Supplementary file 1* Table S6.

## Targeted integration junction fragment analysis in pig fibroblasts

Individual colonies were scored for GFP expression and prepared for PCR by washing with 1X PBS and resuspension in PCR-safe lysis buffer (10 mM Tris-Cl, pH 8.0; 2 mM EDTA; 2.5% (vol/vol) Tween-20; 2.5% (vol/vol) Triton-X 100; 100 µg/mL Proteinase K followed by incubation at 50℃ for 60 min and 95℃ for 15 min. PCR was performed using 1X Accustart Supermix (Quanta) with the primers: 5' junction F-5' TAGAGTCACCCAAGTCCCGT-3', R-5'- ACTGATTGGCCGCTTCTCCT-3'; 3' junction F-5'- GGAGGTGTGGGAGGTTTTT-3', R-5'- TGATTTCATGACTTGCTGGCT-3'. ROSA TALEN sequences are: TAL FNG NI NI HD HD NG NN NI NG NG HD NG NG NN NN; TAL RHD NN NG NI HD NI HD HD NG NN HD NG HD NI NI NG.

## K-592 flow cytometry

K-562 cells were assessed for GFP expression every 7 days for 28 days following electroporation. Flow cytometry was conducted on an LSRII instrument (Becton Dickinson) and data was analyzed using FlowJo software v10 (Becton Dickinson). Dead cells were excluded from analysis by abnormal scatter profile and exclusion based on Sytox Blue viability dye (Thermo Fisher Scientific).

Junction PCR to detect targeted integration was conducted using external genomic primers outside of the 48 bp homology region and internal primers complementary to the expression cassette. PCR was conducted using Accuprime HIFI Taq (Thermo Fisher Scientific). PCR products from bulk population were sequenced directly.

## Quantification and statistical analysis

Statistical analyses were performed using GraphPad Prism software. Data plots represent mean +/- s.e.m. of n independent experiments, indicated in the text. *p* values were calculated with two-tailed unpaired *t*-test. Statistical parameters are included in the Figure legends.

## Data and software availability

Our webtool *GTagHD* was developed to assist users in designing oligonucleotides for targeted integration using the pGTag vector suite (*Mann et al., 2019*). GTagHD guides users through entering: 1) the guide RNA for cutting their cargo-containing plasmid; 2) the guide RNA for cutting their genomic DNA sequence; 3) the genomic DNA sequence, in the form of a GenBank accession number or copy/pasted DNA sequence; and 4) the length of microhomology to be used in integrating the plasmid cargo. If the user is utilizing one of the pGTag series plasmids, GTagHD can also

generate a GenBank/ApE formatted file for that plasmid, which includes the user's incorporated oligonucleotide sequences. GTagHD is freely available online at http://genesculpt.org/gtaghd/ and for download at https://github.com/Dobbs-Lab/GTagHD.

## Acknowledgements

This work was supported by NIH grants R24OD020166 (JJE, MM, DLD, KJC, SCE), GM088424 (JJE), and GM63904 (SCE). Research reported in this publication was made possible in part by the services of the Kansas University Genome Sequencing Core Laboratory supported by the National Institute of General Medical Sciences (NIGMS) of the NIH under award number P20GM103638.

## Additional information

### Competing interests

Wesley A Wierson: Interests in Lifengine and Lifengine Animal Health. Dennis A Webster, Stacy L Solin, Daniel F Carlson: Shares in Recombinetics, Inc. Stephen C Ekker: Shares in Lifengine, and Lifengine Animal Health. Karl J Clark: Shares in Recombinetics, Inc, Lifengine and Lifengine Animal Health. Jeffrey Essner: JJE has a financial conflict of interest with Recombinetics, Inc; Immusoft, Inc; LifEngine and LifEngine Animal Technologies;. The other authors declare that no competing interests exist.

### Funding

| Funder | Grant reference number | Author |
| --- | --- | --- |
| NIH Office of the Director | R24OD020166 | Jeffrey Essner<br>Maura McGrail<br>Drena L Dobbs<br>Karl Clark<br>Stephen C Ekker |
| National Institutes of Health | GM088424 | Jeffrey Essner |
| National Institutes of Health | GM63904 | Stephen C Ekker |

The funders had no role in study design, data collection and interpretation, or the decision to submit the work for publication.

### Author contributions

Wesley A Wierson, Jeffrey Essner, Conceptualization, Resources, Data curation, Formal analysis, Supervision, Funding acquisition, Validation, Investigation, Visualization, Methodology, Writing - original draft, Project administration, Writing - review and editing; Jordan M Welker, Melanie E Torrie, Conceptualization, Resources, Data curation, Formal analysis, Supervision, Investigation, Methodology, Writing - original draft, Project administration; Maira P Almeida, Trevor J Weiss, Conceptualization, Resources, Data curation, Formal analysis, Supervision, Investigation, Visualization, Methodology, Writing - original draft, Project administration; Carla M Mann, Dennis A Webster, Beau R Webber, Branden S Moriarity, Conceptualization, Resources, Data curation, Formal analysis, Investigation, Methodology, Writing - original draft, Project administration; Sekhar Kambakam, Data curation, Formal analysis, Investigation, Methodology; Macy K Vollbrecht, Christopher S Mikelson, Conceptualization, Resources, Data curation, Investigation; Merrina Lan, Zhitao Ming, Jeffrey A Haltom, Data curation, Investigation; Kenna C McKeighan, Conceptualization, Resources, Investigation, Methodology; Jacklyn Levey, Resources, Data curation, Formal analysis, Investigation, Methodology, Writing - original draft; Alec Wehmeier, Data curation, Investigation, Methodology; Kristen M Kwan, Chi-Bin Chien, Darius Balciunas, Karl J Clark, Resources, Investigation, Methodology; Stephen C Ekker, Resources, Investigation; Stacy L Solin, Conceptualization, Resources, Data curation, Investigation, Methodology, Writing - original draft; Daniel F Carlson, Conceptualization, Resources, Formal analysis, Methodology; Drena L Dobbs, Conceptualization, Resources, Methodology; Maura McGrail, Conceptualization, Resources, Data curation, Formal analysis, Supervision, Funding

acquisition, Investigation, Methodology, Writing - original draft, Project administration, Writing - review and editing

### Author ORCIDs

Kristen M Kwan (iD) https://orcid.org/0000-0003-0052-275X
Darius Balciunas (iD) http://orcid.org/0000-0003-1938-3243
Stephen C Ekker (iD) http://orcid.org/0000-0003-0726-4212
Karl J Clark (iD) https://orcid.org/0000-0002-9637-0967
Maura McGrail (iD) http://orcid.org/0000-0001-9308-6189
Jeffrey Essner (iD) https://orcid.org/0000-0001-8816-3848

### Ethics

Animal experimentation: This study was performed in strict accordance with the recommendations in the Guide for the Care and Use of Laboratory Animals of the National Institutes of Health. All of the animals were handled according to approved institutional animal care and use committee (IACUC) protocols (#11-06-6252) of Iowa State University.

### Decision letter and Author response

Decision letter https://doi.org/10.7554/eLife.53968.sa1
Author response https://doi.org/10.7554/eLife.53968.sa2

## Additional files

### Supplementary files

• Supplementary file 1. Supplementary Tables S1 -S7. Supplementary Table S1 sgRNA efficiency. Supplementary Table S2 F0 embryo targeting averages. Supplementary Table S3 F0 embryo targeting data. Supplementary Table S4 F0 Germline transmission by gene. Supplementary Table S5 Individual F0 transmission data. Supplementary Table S6 sgRNA Target site information. Supplementary Table S7 Primer sequences.

• Transparent reporting form

### Data availability

All data are included in this manuscript. MiSeq data are available at https://doi.org/10.5061/dryad. m63xsj3zc.

The following dataset was generated:

| Author(s) | Year | Dataset title | Dataset URL | Database and Identifier |
| --- | --- | --- | --- | --- |
| Essner J, McGrail M | 2020 | Data from: Efficient targeted integration directed by short homology in zebrafish and mammalian cells | https://doi.org/10.5061/dryad.m63xsj3zc | Dryad Digital Repository, 10.5061/dryad.m63xsj3zc |

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
