## [Decision Letter]

**Acceptance summary:**

Wierson et al., report methodological improvement for targeted integration of DNA reporter sequences in zebrafish and mammalian cells. This approach combines the use of short homology arms with the deployment of universal CRISPR sgRNA sequences flanking reporter sequences that allows to expose homology arms in vivo. Along with the suite of vectors that facilitate adopting these approaches this work will stimulate genome engineering-based lines of research.

**Decision letter after peer review:**

Thank you for submitting your article "Efficient targeted integration directed by short homology in zebrafish and mammalian cells" for consideration by *eLife*. Your article has been reviewed by three peer reviewers, including Lilianna Solnica-Krezel as the Reviewing Editor and Reviewer #1, and the evaluation has been overseen by Richard White as the Senior Editor.

The reviewers have discussed the reviews with one another and the Reviewing Editor has drafted this decision to help you prepare a revised submission.

Summary:

This manuscript reports the engineering of a homology directed repair platform for tagging loci after double strand breaks in zebrafish, pig and human cell lines. The authors demonstrate that short homology arms are ideal for targeting, with longer homology arms offering little advantage. Moreover, in vivo release of a double stranded donor with short homology arms is sufficient to drive targeted homology end joining (HMEJ) repair at high efficiency. The manuscript also reports a workaround for loci that are lowly expressed and thus difficult or impossible to directly screen for successful integration events by fluorescence: inserting a Gal4 instead of a fluorescent protein amplifies the signal in a UAS-FP zebrafish line and thus enables screening by fluorescence. The most interesting aspect of the work was the observation that 5' or 3' homology may compete for one another to reduce efficiency and possibly reduce for precise repair at these 5' and 3' ends.

Despite the fact that there are several previous published studies that have reported alternative approaches for achieving this task, precise gene editing is still a non-routine task for zebrafish labs due to low efficiency and imprecise integration. Thus, the combination of technological advancements in this area and the tools generated here that can be broadly shared will have a major impact on the zebrafish community as a whole, and also advance gene editing approaches in mammalian cells.

Therefore, although there was a discussion about the novelty and advance reported, the reviewers consider the manuscript in principle suitable for publication in *eLife*. However, they have raised a number of questions and concerns that will need to be addressed before the manuscript can be published.

Essential revisions:

1) The Southern blots shown in Figure 4B, C should include images of higher molecular weight regions, to more fully assess potential off-targets integrations. Currently they are limited to ca. 8Kb.

2) The experiments presented in Figure 2F for *cx43.4* support the notion that when the Universal CRISPR sgRNA and CRISPR/Cas9 are co-injected with the donor construct to expose homology arms in vivo, increasing the length of homology arms to 1kb does not increase efficiency of integration based on reporter expression in F0. However, the experiments comparing the circular and in vitro linearized construct with the same homology arms of 1kb require providing further detail and discussion.

In particular, it is not clear where in the donor construct was the cut introduced to linearize the HR construct? The in vitro linearized constructs appear to have overhangs that are non-homologous to the target locus (based on what could be deduced from the Materials and methods). Shin et al., 2014 carried systematic comparison of integration efficiency of circular or in vitro linearized at various positions, HR constructs, concluding that internal cuts within homology arms increase efficiency over a circular donor. The current work that introduces cuts in vivo and exposes homology arms is consistent with these results. Therefore, assessing the effect of different cut sites as well as in vitro/in vivo generation of linear ends will be important to clarify what works best. This can be done by assessing reporter expression in F0, without studying germline transmission.

3) An inherent problem of all studies published on CRISPR/Cas9 gene editing technologies: in the absence of data providing direct cross-comparisons of different strategies it is difficult to compare the reported efficiencies across different approaches/research groups, and thus not possible to judge for a reviewer (or reader) whether this technique will really be superior compared to others. In this manuscript, the authors do provide some level of comparison by comparing with uncut and long homology arms and mention other studies in their discussion. However, this should be extended. For example, the study by Luo et al., 2018, appears overall very similar apart from targeting the last exon. If the same gRNAs have been used in the past by other reports, a direct comparison of efficiencies would be helpful.

4) A general comment on how well your genome targeted CRISPR or TALEN reagents worked to target double strand break at each locus will be helpful. I feel many do not grasp how important the genome targeted double strand break is for HDR/HMEJ. Please add in these data and highlight in the Results.

5) Subsection “Dual homology arm liberation directs precise 5’ and 3’ integration in somatic tissue”: This reads like a formal observation of multiple highly speculative molecular events. Please revise accordingly.

[Editors' note: further revisions were suggested prior to acceptance, as described below.]

Thank you for submitting your article "Efficient targeted integration directed by short homology in zebrafish and mammalian cells" for consideration by *eLife*. Your article has been reviewed by a Reviewing Editor and Richard White as the Senior Editor.

Revisions:

Your revised manuscript largely addressed the reviewers' questions and concerns. The manuscript is suitable for publication when the following textual edits are incorporated.

1) The Abstract and Discussion should reflect the changes made in response to reviewers. Please, revise the following sentences in the Abstract: "Here, we describe a set of resources to streamline precision gene targeting in zebrafish", to "Here, we describe a set of resources to streamline reporter gene knock-ins in zebrafish".

This is important distinction, given that the Abstracts also states "1 kb long homology arms did not increase targeting efficiency." However, it remains unclear whether this is the case for both reporter gene knock-ins and generation of in-frame fusion proteins in endogenous genes. Indeed, as noted by the reviewers, universal precision genome editing requires a higher bar of demonstrating the ability to generate protein fusions. The authors cite in their letter "manuscript" in preparation and note "Our preliminary studies show the frequency of precision integration is reduced with these vectors, however, precise integration events can be recovered when screening higher numbers of F0s. We are assembling all of our data to provide a rigorous analysis of these observations prior to publication." Therefore, this rigorous analysis of precise integration events for generating protein fusions is needed before the efficiencies of short and long homology arms for this endpoint can be compared.

2) Therefore, the above considerations should be added to the Discussion following the section comparing the short and long homology arms efficiency.

3) The major and exciting advance of the approach is the combination of using short homology arms AND releasing short homology arms of the injected donor constructs in vivo, but this is not clearly stated in the Abstract. Please consider revising the following sentence "Our vector series, pGTag (plasmids for Gene Tagging), contains reporters flanked by a universal CRISPR sgRNA sequence to target double strand breaks in vivo and expose homology arms” "Our vector series, pGTag (plasmids for Gene Tagging), contains reporters flanked by a universal CRISPR sgRNA sequence enabling to expose in vivo homology arms and target double strand breaks."

---

## [Author Response]

Essential revisions:1) The Southern blots shown in Figure 4B, C should include images of higher molecular weight regions, to more fully assess potential off-targets integrations. Currently they are limited to ca. 8Kb.

In the revised manuscript, in Figure 4 we used the entire image of the blots captured with our BioRad imaging system. Additional bands indicating off-targeting in the *tyr* and *noto* integration lines were not observed. However, we have previously observed additional bands using our short homology arm targeted integration strategy to integrate a reporter at *noto* as described in Wierson et al., 2019. This is now noted in the Results.

2) The experiments presented in Figure 2F for cx43.4 support the notion that when the Universal CRISPR sgRNA and CRISPR/Cas9 are co-injected with the donor construct to expose homology arms in vivo, increasing the length of homology arms to 1kb does not increase efficiency of integration based on reporter expression in F0. However, the experiments comparing the circular and in vitro linearized construct with the same homology arms of 1kb require providing further detail and discussion.In particular, it is not clear where in the donor construct was the cut introduced to linearize the HR construct?

We added this detail to the Results and Materials and methods. We thank the reviewers for noting this!

The in vitro linearized constructs appear to have overhangs that are non-homologous to the target locus (based on what could be deduced from the Materials and methods). Shin et al., 2014 carried systematic comparison of integration efficiency of circular or in vitro linearized at various positions, HR constructs, concluding that internal cuts within homology arms increase efficiency over a circular donor. The current work that introduces cuts in vivo and exposes homology arms is consistent with these results. Therefore, assessing the effect of different cut sites as well as in vitro/in vivo generation of linear ends will be important to clarify what works best. This can be done by assessing reporter expression in F0, without studying germline transmission.

We expanded the Materials and methods to clarify that the experiments comparing long vs. short homology arms for *cx43.4* used unique cut sites in the homology arms to create a linear DNA fragment. The homology extends to the end of the linear targeting construct, similar to Shin et al., 2014. This has also been explained in the Results of our revised manuscript. The *esama* linear DNA fragment did contain non-homologous sequences. However, in vivo liberation of homology arms by CRISPR/Cas9 cleavage leaves short segments of non-homologous sequence at the ends of the homology arms. We addressed how this may impact on-target integration in the discussion.

3) An inherent problem of all studies published on CRISPR/Cas9 gene editing technologies: in the absence of data providing direct cross-comparisons of different strategies it is difficult to compare the reported efficiencies across different approaches/research groups, and thus not possible to judge for a reviewer (or reader) whether this technique will really be superior compared to others. In this manuscript, the authors do provide some level of comparison by comparing with uncut and long homology arms and mention other studies in their discussion. However, this should be extended. For example, the study by Luo et al., 2018, appears overall very similar apart from targeting the last exon. If the same gRNAs have been used in the past by other reports, a direct comparison of efficiencies would be helpful.

We apologize for the omission of the Luo et al., 2018 reference and have added it to the Discussion. To our knowledge, none of the sgRNAs used in this study have been used previously.

4) A general comment on how well your genome targeted CRISPR or TALEN reagents worked to target double strand break at each locus will be helpful. I feel many do not grasp how important the genome targeted double strand break is for HDR/HMEJ. Please add in these data and highlight in the Results.

To address this point, we added MiSeq and ICE analysis of indel frequency at the genomic target sites for the sgRNAs used for targeted integration (Supplementary file 1). The mutagenesis efficiency of most sgRNAs is at least 80%. We have highlighted the efficiencies in the Results.

5) Subsection “Dual homology arm liberation directs precise 5’ and 3’ integration in somatic tissue”: This reads like a formal observation of multiple highly speculative molecular events. Please revise accordingly.

We removed the simultaneous emphasis in these sentences and softened the language by adding “likely”.

[Editors' note: further revisions were suggested prior to acceptance, as described below.]

Revisions:Your revised manuscript largely addressed the reviewers' questions and concerns. The manuscript is suitable for publication when the following textual edits are incorporated.1) The Abstract and Discussion should reflect the changes made in response to reviewers. Please, revise the following sentences in the Abstract: "Here, we describe a set of resources to streamline precision gene targeting in zebrafish", to "Here, we describe a set of resources to streamline reporter gene knock-ins in zebrafish".

We have made the substitution.

This is important distinction, given that the Abstracts also states "1 kb long homology arms did not increase targeting efficiency." However, it remains unclear whether this is the case for both reporter gene knock-ins and generation of in-frame fusion proteins in endogenous genes. Indeed, as noted by the reviewers, universal precision genome editing requires a higher bar of demonstrating the ability to generate protein fusions. The authors cite in their letter "manuscript" in preparation and note "Our preliminary studies show the frequency of precision integration is reduced with these vectors, however, precise integration events can be recovered when screening higher numbers of F0s. We are assembling all of our data to provide a rigorous analysis of these observations prior to publication." Therefore, this rigorous analysis of precise integration events for generating protein fusions is needed before the efficiencies of short and long homology arms for this endpoint can be compared.

We have removed the sentence "1 kb long homology arms did not increase targeting efficiency" from the Abstract, as this could only be assessed at one locus. However, in the current manuscript we used a 2A peptide which requires the same in frame rigor imposed by making a protein fusion, i.e. the reporter gene is required to be in the same reading frame as the targeted gene. We do believe that this approach along with our junction fragment analysis is rigorous for the short homology arms, just not the comparisons between longer and shorter homology arms. We have added the following sentence to emphasize this in the Results and Discussion:

In the Results: “For fluorescence to be detected, the integrated reported gene is required to be in frame with the open reading frame of *noto*.”

In the Discussion: “However, further experiments are required at additional loci to determine whether the advantages of using short homology in targeting vectors represents a universal rule.”

2) Therefore, the above considerations should be added to the Discussion following the section comparing the short and long homology arms efficiency.

See above.

3) The major and exciting advance of the approach is the combination of using short homology arms AND releasing short homology arms of the injected donor constructs in vivo, but this is not clearly stated in the Abstract. Please consider revising the following sentence "Our vector series, pGTag (plasmids for Gene Tagging), contains reporters flanked by a universal CRISPR sgRNA sequence to target double strand breaks in vivo and expose homology arms” "Our vector series, pGTag (plasmids for Gene Tagging), contains reporters flanked by a universal CRISPR sgRNA sequence enabling to expose in vivo homology arms and target double strand breaks."

Thank you for the suggestion. We have incorporated the following sentence “Our vector series, pGTag (plasmids for Gene Tagging), contains reporters flanked by a universal CRISPR sgRNA sequence which enables in vivo exposure of the homology arms.”